



# Bayesian analysis of early warning signals using a time-dependent model

Eirik Myrvoll-Nilsen[1], Luc Hallali[1], and Martin Wibe Rypdal[1]

[1]Department of Mathematics and Statistics, UiT The Arctic University of Norway, N-9037 Tromsø, Norway

**Correspondence:** Eirik Myrvoll-Nilsen (eirik.myrvoll-nilsen@uit.no)

**Abstract.**

A tipping point is defined by the IPCC as a critical threshold beyond which a system reorganizes, often abruptly and/or irreversibly. Tipping points can be crossed solely by internal variation in the system or by approaching a bifurcation point where the current state loses stability and forces the system to move to another stable state. It is currently debated whether or not Dansgaard-Oeschger (DO) events, abrupt warmings occurring during the last glacial period, are noise-induced or caused by the system reaching a bifurcation point. It can be shown that before a bifurcation point is reached there are observable changes in the statistical properties of the state variable. These are known as early warning signals and include increased fluctuation and correlation time. To express this behaviour we propose a new model based on the well-known first order autoregressive process (AR), with modifications to the correlation parameter such that it depends linearly on time. In order to estimate the time evolution of the correlation parameter we adopt a hierarchical Bayesian modeling framework, from which Bayesian analysis can be performed using the methodology of integrated nested Laplace approximations. We then apply the model to segments of the oxygen isotope ratios from the Northern Greenland Ice Core Project record corresponding to 17 DO events. Early warning signals were detected and found statistically significant for a number of DO events, suggesting that such events could indeed be caused by approaching a bifurcation point. The methodology developed to perform the given early warning analyses can be applied more generally, and is publicly available as the R-package `INLA.ews`.

## 1 Introduction

An equilibrium state is said to be stable if the system returns to the same state following a small perturbation in any direction. If the state of a component of the climate system, by crossing some threshold in the form of an unstable barrier separating two basins of attraction, changes from one stable equilibrium to another it is said to have reached a tipping point. Components of the Earth system has experienced tipping points numerous times in the past, leading to abrupt transitions in the climate system. These transitions are well documented in paleoclimatic proxy records. Notably, in Greenland ice core records of oxygen isotope ratios ($\delta^{18}$O) and dust concentrations there is evidence that large and abrupt climatic transitions from Greenland stadial (GS) to Greenland interstadial (GI) conditions took place in the last glacial interval (110,000–12,000 years before 2000 AD, hereafter denoted yr b2k). These are known as Dansgaard-Oeschger (DO) events (Dansgaard et al., 1984, 1993) and are characterized by cycles where the temperature increased substantially, up to 16.5°C for single events, over the course of a few decades followed





by a more gradual cooling, over centuries to millenia, back to the GS state. A total of 17 DO events (Svensson et al., 2008) have been found for the past 60 kyr before present (BP) and they represent some of the most pronounced examples of abrupt transitions in past climate observed in paleoclimatic records.

It is widely accepted that such transitions are associated with a change in the meriodional overturning circulation (MOC)
(Bond et al., 1999; Li et al., 2010) causing a loss of sea ice in the North Atlantic. However, the physical mechanisms that caused these changes in the MOC and how they triggered DO events are less understood. Some studies have found that DO events exhibit a periodicity of 1470 years (Schulz, 2002), which have made some scientists suggest that the events have been triggered by changes in the earth system caused by changing solar forcing (Braun et al., 2005). Others suggest that the transitions have been triggered by random fluctuations in the Earth system, without any significant changes to the underlying system caused by
external forcing (Ditlevsen et al., 2007). Treating the GS and GI states as stable equilibria in a dynamical system representing the Greenland climate, and studying the statistical behaviour related to the stability of the system in the period preceeding DO events, can help determine whether or not they are forced or random and thus possibly constrain the number of plausible physical causes that trigger the events.

The behaviour around a tipping point can be analyzed by expressing the changes of the state-variable using a potential,
wherein valleys represent the basins of attraction that are separated by an unstable fixed point. If the tipping point is reached solely from perturbations caused by internal variation of the system, then it is said to be noise-induced. However, if the dynamics of the system depend on some slowly varying control parameter the equilibrium points may shift, vanish or spawn as a function of the control parameter. This means that the stability of a fixed point can change over time and eventually be lost, making the system move to another equilibrium. Points in the control parameter space for which the qualitative behaviour of a
system changes, e.g. change in stability or the number of fixed points, are called bifurcation points, and tipping points caused by the control parameter crossing a bifurcation point are said to be bifurcation-induced.

By assuming that a time-dependent state-variable $x(t)$, representing for example the $\delta^{18}O$ ratio, vary over some potential $V(x)$ with stochastic forcing corresponding to a white noise process $dB(t)$, expressed as the derivative of a Brownian motion, then the stability of the system can be modeled using the stochastic differential equation

$$dx(t) = F(x(t))dt + \sigma dB(t). \tag{1}$$

One could interpret this equation as describing the motion of some particle in the presence of a potential $V(x)$, with drift expressed by $F(x) = -V'(x)$ and a diffusion term $\sigma dB(t)$ describing the noise that acts on the particle.

Take for example the cusp catastrophe model where the potential is given by

$$V(x, \mu, \xi) = \frac{x^4}{4} - \xi \frac{x^2}{2} - \mu x, \tag{2}$$

where $\mu(t)$ is a slowly changing control parameter and $\xi$ is a shape parameter that we in this example set equal to $\xi = 1$. The change in position $dx(t)$ at some time $t$ is then given by

$$dx(t) = -x^3 + \xi x + \mu + \sigma dB(t). \tag{3}$$





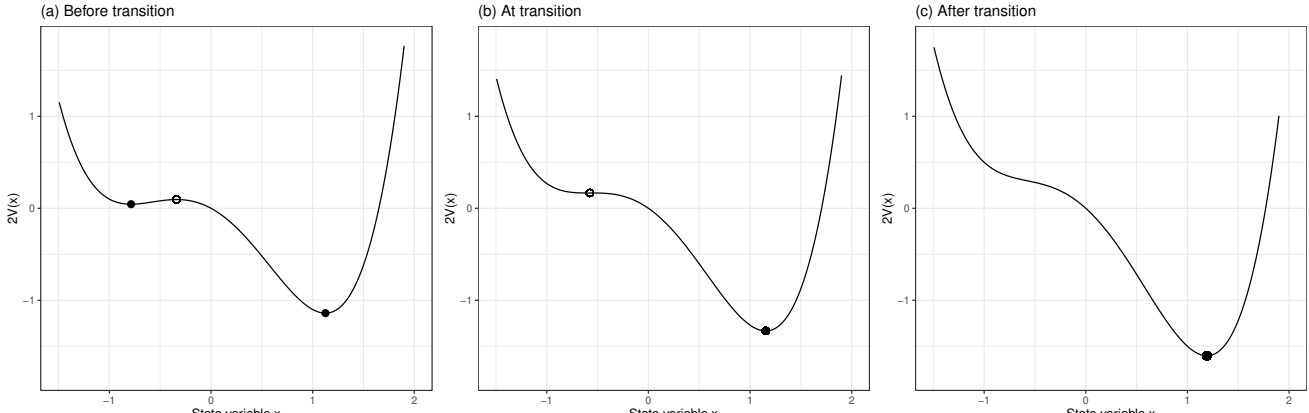

**Figure 1.** The potential over the set of state variables before, at and after the control parameter has reached the bifurcation point $\mu_2$. Panel (a) shows the potential and fixed points for some $\mu < \mu_2$, and panels (b)–(c) shows the same for $\mu = \mu_2$ and $\mu > \mu_2$, respectively. When the control parameter approaches the bifurcation point $\mu_2$, the stability of the stable fixed point $x_1$ decreases and eventually collapses at $x_1 = x_2 = -\sqrt{\xi/3}$, leaving $x_3$ as the only (stable) fixed point.

It can be shown that the bifurcation points are

$$\mu_1 = -\frac{2}{3}\sqrt{\xi^3/3} \quad \text{and} \quad \mu_2 = \frac{2}{3}\sqrt{\xi^3/3}. \tag{4}$$

Crossing the bifurcation points changes the number of fixed points. For $\mu_1 < \mu < \mu_2$ there are two stable fixed points and one unstable, and for $\mu < \mu_1$ or $\mu > \mu_2$ there is only one (stable) fixed point. The change of stability can be depicted by plotting the potential before and after the bifurcation points, see Fig. 1 for an illustration where the control variable varies around $\mu_2$. The change in values and stability of the fixed points as we increase the control parameter is illustrated in the bifurcation diagram Fig. 2, which include the stable fixed points $x_1$ (lower solid curve) and $x_3$ (upper solid curve) and the unstable fixed points

$x_2$ (middle dashed curve), representing the separating barrier. The diagram also includes a simulated process generated by the same potential which demonstrates how abruptly the state variable changes when the system crosses the tipping threshold $x_2$, which happens before the control parameter reaches the bifurcation point $\mu_2$ due to the diffusion term $\sigma dB(t)$.

The nature of an equilibrium can be investigated by examining the linear approximation in its nearby domain. Linearizing (1) around some stable fixed point $x_s$ yields

$$dx(t) = -\lambda(x(t) - x_s)dt + \sigma dB(t), \tag{5}$$

where $\lambda = -F'(x_s)$. This is known as the Langevin stochastic differential equation and has the solution

$$x(t) = x_0 + \int_{-\infty}^{t} g(t - s)dB(s), \tag{6}$$




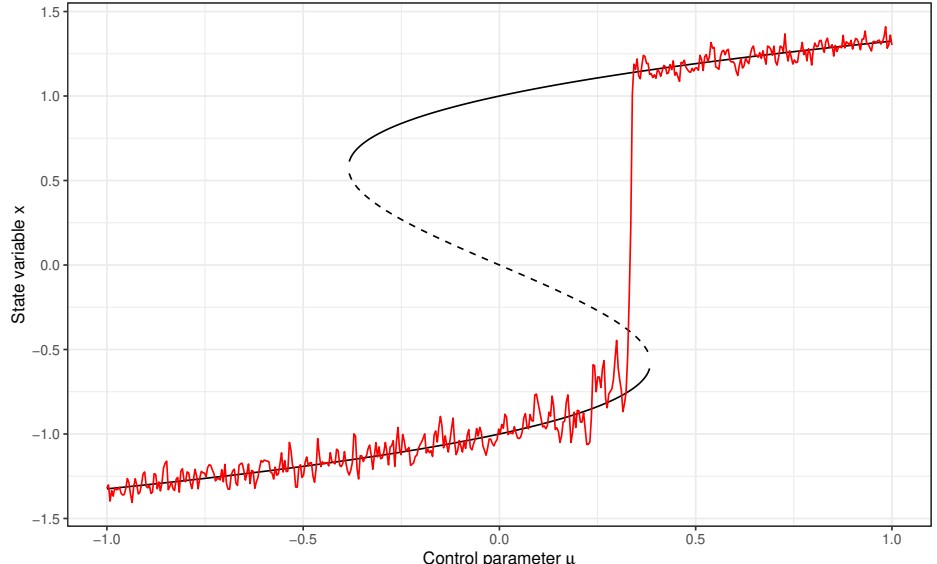

**Figure 2.** The bifurcation diagram of the cusp catastrophe model. The black curve represent the fixed points of the state variable $x$ given the changing control parameter $\mu \in (-1, 1)$. The solid curves represent stable fixed points $x_1$ and $x_3$, and the dashed curve represent unstable fixed points $x_2$. The red line represent a simulation of Eq. (3) with $\sigma = 0.2$. As the control parameter $\mu$ approaches the bifurcation point $\mu_2$ the stability of $x_1$ decreases which is expressed by increased variance and correlation in the simulated process, causing the system to cross the tipping point $x_2$ prematurely.

with Green's function

$$g(t) = \begin{cases} \exp(-\lambda t), & x \geq 0 \\ 0, & x < 0 \end{cases}. \tag{7}$$

This solution forms an Ornstein-Uhlenbeck (OU) process, which under discretization is a first order autoregressive (AR) process with variance $\mathrm{Var}(x_t) = \sigma^2/(2\lambda)$ and lag-one autocorrelation parameter $\phi(t) = \exp(-\lambda)$.

$$x_t = \phi x_{t-1} + \varepsilon_t, \qquad \varepsilon \sim \mathcal{N}\left(0, \frac{1 - \phi^2}{2\lambda}\sigma^2\right) \tag{8}$$

When the control parameter approaches a bifurcation point we expect increased variance and correlation, as could be observed in Fig. 2. These changes in statistical behaviour are called early-warning signals (EWS) of the bifurcation point, or critical slowing down (Lenton et al., 2012; Dakos et al., 2008), and can be used as precursors to help determine whether or not a tipping point is imminent. In fact, recent studies have discovered that more components in the earth system exhibit EWS and are at risk of approaching or have already reached a tipping point. This include the western Greenland ice sheets (Boers and Rypdal, 2021), the Atlantic meridional overturning circulation (Boers, 2021) and the Amazon rainforest (Boulton et al., 2022).

Analysis of EWS for DO events in the high-dimensional Greenland ice core record has been conducted by others, e.g. Ditlevsen and Johnsen (2010) whom applied a Monte Carlo approach to detect increased variance and autocorrelation in a



system driven by white noise. Under these assumptions they were unable to detect a statistical significant increase in EWS suggesting that DO events are noise-induced. However, using different model assumptions, Rypdal (2016) was able to detect statistically significant EWS in an ensemble of DO events. This was achieved by analyzing individual frequency bands separately, using a fractional Gaussian noise (fGn) (Mandelbrot and Van Ness, 1968) model to describe the noise. Fractional
Gaussian noise is a long-range dependent model for which the Green's function in (6) is scale-invariant

$$
g(t) = \begin{cases} t^{H-3/2}, & x \geq 0 \\ 0, & x < 0 \end{cases}.
\tag{9}
$$

$H \in (0.5, 1)$ is the memory coefficient known as the Hurst exponent. Fractional Gaussian noise have been shown to be more realistic for describing components in the Earth system where the power spectrum does not follow an exponential decay, such as monthly to centennial global and local mean surface temperature data (Lovejoy and Schertzer, 2013; Huybers and Curry,
2006; Rybski et al., 2006; Rypdal and Rypdal, 2016; Franzke et al., 2015; Fredriksen and Rypdal, 2016; Løvsletten and Rypdal, 2016; Myrvoll-Nilsen et al., 2019). Rypdal (2016) was able to detect an increase of variance of the high-frequency fluctuations for the ensemble average of the 17 DO events at a 5% significance level, and individually for five separate events. These results were corroborated by Boers (2018) whom applied a similar strategy to the higher resolution of the NGRIP $\delta^{18}$O data set (Andersen et al., 2004; Gkinis et al., 2014) on which he applied interpolation to obtain time series with regular 5-year sampling
steps.

Most approaches for detecting EWS in the current literature require estimation of statistical properties in a sliding window, e.g. by producing Fourier surrogates and estimating the Kendall's $\tau$ statistic for each iteration. Consequently, this presents a choice on the length of the window. Using a small window will allow for the momentary state to be better depicted, but there will be fewer points used in the estimation hence accuracy will suffer. On the other hand, if a larger window is used
the estimated statistics will be more accurate, but less representative of the momentary state as it represents an average over a larger time scale. The optimal choice of window length should ideally represent a good trade-off between accuracy and ability to represent momentary evolution, but this can be hard to determine in practice. In this paper we circumvent this issue and present a model-based approach where such a compromise is not required. By assuming that the correlation parameter is time-dependent, following a specific linear structure, it is possible to formulate this into a hierarchical Bayesian model for
which well-known computational frameworks can be applied. A Bayesian approach has the additional benefit of providing uncertainty estimates in the form of posterior distributions.

The paper is structured as follows. A description of the data used in this paper is included in section 2. Section 3 details our methodology, including how we treat time-dependence, how to formulate our model as a hierarchical Bayesian model and how to perform statistical inference efficiently. Results are presented in section 4 where our framework is applied first to simulated
data, then to Dansgaard-Oeschger events observed in the $\delta^{18}$O data from the NGRIP record. Our results are compared with those obtained by Ditlevsen and Johnsen (2010), Rypdal (2016) and Boers (2018). Further discussion and conclusions are provided in section 5.



## 2 NGRIP ice core data

The $\delta^{18}$O ratios are frequently used in paleoscience as proxies for temperature of precipitation (Johnsen et al., 1992, 2001;
Dansgaard et al., 1993; Andersen et al., 2004), where higher ratios signals colder climates and, conversely, warmer climates
tend to result in lower ratios. We employ the $\delta^{18}$O proxy record from the Northern Greenland Ice core Project (NGRIP) (North
Greenland Ice Core Project members, 2004; Gkinis et al., 2014; Ruth et al., 2003). There are currently two different versions
of the NGRIP/GICC05 data, at different resolutions. We will apply our methodology to the higher resolution record, which is
sampled every 5cm in depth. The NGRIP $\delta^{18}$O proxy record is defined on a temporal axis given by the Greenland Ice Core
Chronology 2005 (GICC05) (Vinther et al., 2006; Rasmussen et al., 2006; Andersen et al., 2006; Svensson et al., 2008) which
thus pairs the $\delta^{18}$O measurements with a corresponding age, stretching back to 60 kyrs b2k. We use segments of the $\delta^{18}$O
record corresponding to Greenland stadial phases preceding DO onsets, as given by Table 2 of Rasmussen et al. (2014). The
data used in this paper can be downloaded from https://www.iceandclimate.nbi.ku.dk/data/ (last accessed: **day month year**)

## 3 Methodology

During critical slowing down stationarity can no longer be assumed as we expect both the correlation and variance to increase.
For an AR(1) process $\boldsymbol{x} = (x_1, ..., x_n)^\top$ sampled at times $t_1, ..., t_n$, we assume that the increase in correlation can be expressed
by representing the lag-one autocorrelation parameter as a linear function of time

$$\phi(t) = a + bt, \qquad 0 \le t \le 1, \tag{10}$$

where $a$ and $b$ are two unknown parameters. The joint vector of variables $\boldsymbol{x} = (x_1, ..., x_n)^\top$ forms a multivariate Gaussian
process

$$\boldsymbol{x} \sim \mathcal{N}(\mathbf{0}, \boldsymbol{\Sigma}), \tag{11}$$

where the covariance matrix is given by

$$\Sigma_{ij} = \mathrm{Cov}(x_i, x_j). \tag{12}$$

The time-dependent AR(1) process is expressed by the difference equation

$$x_t = \phi(t)x_{t-1} + \varepsilon_t, \qquad \varepsilon \sim \mathcal{N}(0, \sigma_\varepsilon^2), \qquad t = t_1, ..., t_n, \tag{13}$$

for which the covariance between two variables $x_i$ and $x_j$ is given by $\mathrm{Cov}(x_i, x_j)$.

Since the covariance matrix is almost always dense it is computationally beneficial to instead work with the inverse-
covariance matrix, also known as the precision matrix $\boldsymbol{Q} = \boldsymbol{\Sigma}^{-1}$. It can be shown that for a time-dependent AR(1) process





the precision matrix is sparse and equal to

$$
\quad Q = \frac{1}{\sigma^2}
\begin{pmatrix}
1 + \phi(t_2)^2 & -\phi(t_2) & & & \\
-\phi(t_2) & 1 + \phi(t_3)^2 & -\phi(t_3) & & \\
\ddots & \ddots & \ddots & & \\
& -\phi(t_{n-1}) & 1 + \phi(t_n)^2 & -\phi(t_n) \\
& & -\phi(t_n) & 1
\end{pmatrix}. \tag{14}
$$

Gaussian processes with sparse precision matrices are known as Gaussian Markov random fields, and there is a wealth of efficient algorithms for fast Bayesian inference, see e.g. Rue and Held (2005) for a comprehensive discussion on this topic. These computationally efficient properties are not shared by the fractional Gaussian noise for which both the covariance matrix and the precision matrix are dense. This means that essential matrix operations such as computing the Cholesky decomposition

will have a computational cost of $\mathcal{O}(n^3)$ floating point operations (flops), as opposed to $\mathcal{O}(n)$ flops for the AR(1) process. Inference might still be possible to achieve in a reasonable amount of time if the size of the data set remains sufficiently small. For larger data sets, however, both time and memory consumption may become an issue.

In fitting the model it is beneficial that the model parameters are defined on an unconstrained parameter space. We therefore introduce a suitable parameterization for $a$ and $b$ using variations of the logistic transformation. Our reasoning is as follows.

Assuming the lag-one autocorrelation parameter is defined on the interval $(0, 1)$, and since $t \in [0, 1]$, then the slope must be constrained by

$$
|b| < 1, \tag{15}
$$

An unconstrained parameterization for $b$ thus reads

$$
\theta_b = \log\left(\frac{1+b}{1-b}\right) \quad \Longleftrightarrow \quad b = -1 + \frac{2}{1 + \exp(-\theta_b)}, \qquad \theta_b \in (-\infty, \infty). \tag{16}
$$

The parameter space for $a$ depend on the current state of $b$

$$
0 < a + bt < 1 \quad \Longleftrightarrow \quad -bt < a < 1 - bt. \tag{17}
$$

Let

$$
a_{\text{lower}} = -\min(b, 0) \quad \text{and} \quad a_{\text{upper}} = 1 - \max(b, 0), \tag{18}
$$

then an unconstrained parameterization for $a$ is given by

$$
\quad \theta_a = \log\left(\frac{a - a_{\text{lower}}}{a_{\text{upper}} - a}\right) \quad \Longleftrightarrow \quad a = a_{\text{lower}} + \frac{a_{\text{upper}} - a_{\text{lower}}}{1 + \exp(-\theta_a)}, \qquad \theta_a \in (-\infty, \infty). \tag{19}
$$

### 3.1 Latent Gaussian model formulation

In this paper we adopt a Bayesian framework to estimate the model parameters. This means that parameters are treated as stochastic variables for which prior knowledge, expressed using prior distributions, is incorporated and updated by the likelihood of the observations using Bayes' theorem. Bayesian inference can be obtained by expressing our model as a hierarchical





Bayesian model, wherein the observed state variables are modeled in terms of a random predictor

$$\boldsymbol{\eta} = \beta_0 + \sum_{i=1}^{n_\beta} \beta_i \boldsymbol{z}_i + \boldsymbol{\varepsilon}(\boldsymbol{\theta}) = \boldsymbol{\mu}(\boldsymbol{\beta}) + \boldsymbol{\varepsilon}(\boldsymbol{\theta}). \tag{20}$$

Here, $\beta_0$ represent an intercept, $\beta_i$ are fixed effects corresponding to covariates $\boldsymbol{z}_i$ and $\boldsymbol{\varepsilon}$ are random effects representing some time-dependent noise that depend on some parameters $\boldsymbol{\theta}$. Notably, in the Bayesian framework fixed effects are treated as stochastic variables and must be assigned prior distributions. If the data is already detrended then $\boldsymbol{\mu} = \mathbf{0}$. The covariance

structure of the different components in the model are expressed by a latent field of random variables containing the predictor and all stochastic terms therein , i.e. $\boldsymbol{x} = (\boldsymbol{\eta}, \boldsymbol{\beta}, \boldsymbol{\varepsilon})$. Assigning a Gaussian prior on $\boldsymbol{x}$ the model becomes a latent Gaussian model, a subset of Bayesian hierarchical models for which there exists additional computational frameworks. The latent Gaussian model is specified in three stages as follows.

     The first stage is to specify the likelihood of the model. We assume the likelihood to be conditionally independent given the

latent field $\boldsymbol{x}$, and expressed by a Gaussian distribution with some small negligible fixed variance $\sigma_y^2 \approx 0$ and mean equal to the predictor

$$\boldsymbol{y} \mid \boldsymbol{x} \sim \prod_{i=1}^{n} \mathcal{N}(\eta_i, \sigma_y^2). \tag{21}$$

     The second stage in specifying a latent Gaussian model is to specify a Gaussian prior distribution for the latent field $\boldsymbol{x}$, with mean vector $\boldsymbol{\mu} = \mathrm{E}(\boldsymbol{\theta})$ and precision matrix $\boldsymbol{Q}$. This may depend on some unknown hyperparameters $\boldsymbol{\theta}$ and expresses the

covariance structure of the latent variables. $\boldsymbol{\beta}$ are assigned vague Gaussian priors and the noise term Specifically, for the linear predictor we assume

$$\boldsymbol{x} \mid \boldsymbol{\theta} \sim \mathcal{N}(\boldsymbol{\mu}, \boldsymbol{Q}(\boldsymbol{\theta})^{-1}), \tag{22}$$

such that the latent variables corresponding to a potential $\boldsymbol{\beta}$ component represent vague Gaussian priors and those corresponding to $\boldsymbol{\varepsilon}$ represent the chosen model. The precision matrix is given by Eq. (14).

The final stage concerns the prior distributions of the model parameters, which we assign independently

$$\boldsymbol{\theta} \sim \pi(\kappa)\pi(\theta_a)\pi(\theta_b). \tag{23}$$

For the analysis performed in this study we have assigned a penalised complexity prior (Simpson et al., 2017) for the scaling parameter $\kappa = 1/\sigma^2$ and Gaussian priors for the parameterized memory parameters $\theta_a$ and $\theta_b$.

## 3.2   Inference

In the Bayesian paradigm inference is expressed by the posterior distribution which provides a complete description of the probabilistic nature of the model parameters and latent variables. The joint posterior distribution can be found relatively easily by

$$\pi(\boldsymbol{x}, \boldsymbol{\theta} \mid \boldsymbol{y}) \propto \pi(\boldsymbol{\theta})\pi(\boldsymbol{x} \mid \boldsymbol{\theta}) \prod_{i=1}^{n} \pi(y_i \mid \boldsymbol{x}). \tag{24}$$




We want to estimate the marginal posterior distribution for all hyperparameters and latent variables. These are computed by evaluating the integrals

$$\pi(x_i \mid \boldsymbol{y}) = \int \pi(x_i \mid \boldsymbol{\theta}, \boldsymbol{y})\pi(\boldsymbol{\theta} \mid \boldsymbol{y})d\boldsymbol{\theta} \tag{25}$$

$$\pi(\theta_j \mid \boldsymbol{y}) = \int \pi(\boldsymbol{\theta} \mid \boldsymbol{y})d\boldsymbol{\theta}_{-j}. \tag{26}$$

These integrals are often impossible to evaluate analytically and are typically computed numerically using Markov chain Monte Carlo approaches (Robert et al., 1999). However, these can sometimes be very time consuming for hierarchical models. For latent Gaussian models with a sparse precision matrix there exists a computationally superior alternative in using integrated nested Laplace approximations (INLA) (Rue et al., 2009, 2017). Instead of using simulations, INLA use various numerical optimization techniques to compute an accurate approximation of the posterior marginal distributions. Most importantly is the Laplace approximation (Tierney and Kadane, 1986), which is used to approximate the joint posterior distribution

$$\pi(\boldsymbol{\theta} \mid \boldsymbol{y}) \approx \left. \frac{\pi(\boldsymbol{x}, \boldsymbol{\theta}, \boldsymbol{y})}{\pi_G(\boldsymbol{x} \mid \boldsymbol{\theta}, \boldsymbol{y})} \right|_{\boldsymbol{x}=\boldsymbol{x}^*(\boldsymbol{\theta})}, \tag{27}$$

where $\boldsymbol{x}^*(\boldsymbol{\theta})$ is the mode of the latent field $\boldsymbol{x}(\boldsymbol{\theta})$ and $\pi_G(\boldsymbol{x} \mid \boldsymbol{\theta}, \boldsymbol{y})$ is the Gaussian approximation of

$$\pi(\boldsymbol{x} \mid \boldsymbol{\theta}, \boldsymbol{y}) \propto \pi(\boldsymbol{x} \mid \boldsymbol{\theta})\pi(\boldsymbol{y} \mid \boldsymbol{x}, \boldsymbol{\theta}). \tag{28}$$

The methodology is available as the open source R package `R-INLA`, which can be downloaded at www.r-inla.org (last access: **day month year**).

As there are currently no model components already implemented for `R-INLA` that meet our specifications we are required to implement the model components ourselves using the custom modeling framework of R-INLA called `rgeneric`. This adds more work and complexity in implementing our model, and adds an additional barrier to further adoption of our methodology. To increase accessibility we have implemented the code and made it available as a user-friendly R-package titled `INLA.ews`, available at www.github.com/eirikmn/INLA.ews (last access: **day month year**). Inference can then be produced by executing `inla.ews(y, formula=formula)`, where `y` is a numeric vector containing the data and `formula` describes the trends included in the model. A demonstration of the `INLA.ews` package applied to simulated data can be found in Appendix B, and a detailed description of its features can be found in its accompanying documentation.

### 3.3 Non-constant time steps

To allow for non-constant time steps $\Delta t_k = t_k - t_{k-1}$ we assume

$$\phi(t_k) = e^{-\lambda(t_k)\Delta t_k/c}, \tag{29}$$

where

$$\lambda(t_k) = -\log(a + bt_k), \tag{30}$$





$c = \sum_{k=2}^{n} \Delta t_k/(n-1)$ and $t_k$ has been normalized such that $t_k \in (0,1)$. This modification guarantees that $\phi(t) \longrightarrow 1$ as $\Delta t_k \longrightarrow 0$ and $\phi(t) \longrightarrow 0$ as $\Delta t_k \longrightarrow \infty$. It also ensures that $\phi(t_1) = a$ and $\phi(t_n) = a+b$ which makes the interpretability of the parameters easier.

If we denote $\sigma(t_k)^2 = \sigma^2/(2\lambda(t_k))$, and assume

$$x_1 \sim \mathcal{N}\left(0, \sigma(t_1)^2\right), \tag{31}$$

then the precision matrix for non-constant time steps yields

$$Q = \begin{pmatrix} \left(\frac{1}{\sigma(t_1)^2} + \frac{\phi(t_2)^2}{\sigma(t_2)^2}\right) & -\frac{\phi(t_2)}{\sigma(t_2)^2} & & \\ -\frac{\phi(t_2)}{\sigma(t_2)^2} & \left(\frac{1}{\sigma(t_2)^2} + \frac{\phi(t_3)^2}{\sigma_3^2}\right) & -\frac{\phi(t_3)}{\sigma(t_3)^2} & \\ & \ddots & \ddots & \ddots \\ & -\frac{\phi(t_{n-1})}{\sigma(t_{n-1})^2} & \left(\frac{1}{\sigma(t_{n-1})^2} + \frac{\phi(t_n)^2}{\sigma(t_n)^2}\right) & -\frac{\phi(t_n)}{\sigma(t_n)^2} \\ & & -\frac{\phi(t_n)}{\sigma(t_n)^2} & \frac{1}{\sigma(t_n)^2} \end{pmatrix}.$$

Non-constant time steps can be specified in the `inla.ews` function by using the `timesteps` input argument.

## 235    3.4   Incorporating forcing

Climate components may also be affected by forcing. How the observed component responds to such forcing will be influenced by time-dependence. In this subsection we adopt a similar strategy to **myrvoll-nilsen2020** with changes to allow for time-dependence and non-constant time steps.

Let $F(t)$ denote the known forcing component such that

$$dx(t) = -\lambda\left(x(t) + F(t)\right)dt + dB(t). \tag{32}$$

The model can then be expressed as the sum of two components

$$x(t) = \mu(t) + \varepsilon(t), \tag{33}$$

where $\varepsilon(t)$ is a time-dependent OU process and the forcing response is given by

$$\mu(t) = \sigma_f(t)\int_0^t e^{-\lambda(t)(t-s)}\left(F(s) + F_0\right)ds. \tag{34}$$

$\sigma_f^2(t) = \sigma_f^2/(2\lambda(t))$ is an unknown scaling parameter and $F_0$ is an unknown shift parameter.

Forcing can be incorporated into the model by specifying the `forcing` argument in the `inla.ews` function.





# 4 Results

## 4.1 Accuracy test on simulated data

To test the accuracy and robustness of the time-dependent AR(1) model we fit the model to a number of simulations. Specifi-
cally, we perform accuracy tests using a grid of $b \in [-0.8, 0.8]$ with increments of 0.1, and choose the parameter $a$ correspond-
ing to $\theta_a = 0$. For each $b$ we draw $n_r = 1000$ time series of length $n = 500$ and $n = 1000$ from the time-dependent AR(1)
model. The model is fitted using `R-INLA` with the same specifications as used in the `INLA.ews` package. To quantify the
accuracy of the model we compare the posterior marginal mean of the slope $\hat{b} = \mathrm{E}(\pi(b \mid \boldsymbol{y}))$ to the true values $b$. We also
compute the posterior probability of the slope being positive $P(b > 0)$. Ideally, we want $\hat{b}$ to be as close to $b$ as possible, and
$P(b > 0) > 0.5$ if $b > 0$ and, conversely, $P(b > 0) < 0.5$ if $b < 0$.

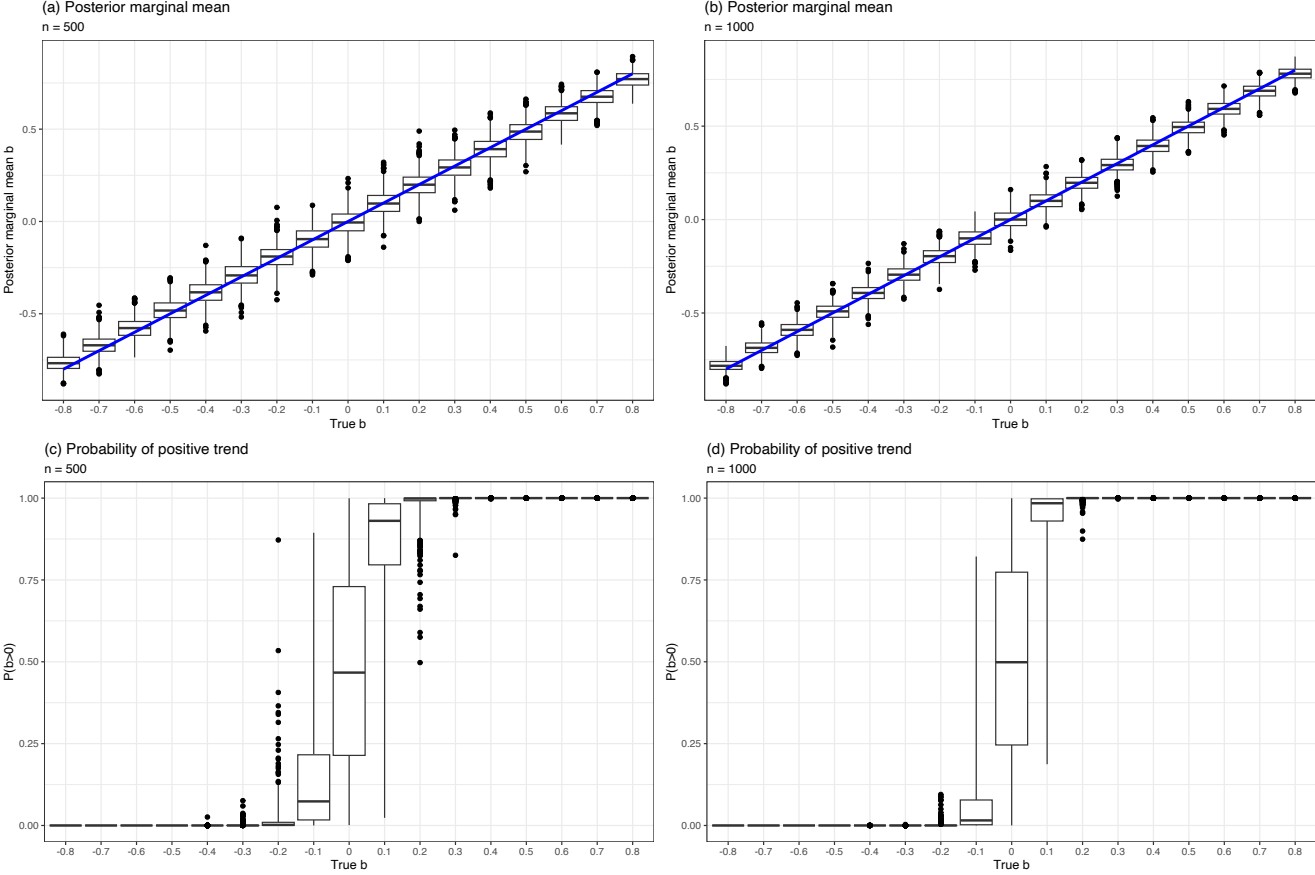

**Figure 3.** Box plots representing the results of the accuracy test for $n_r = 1000$ simulated time series of length $n = 500$ for each
$b \in [-0.8, 0.8]$. Panels (a) and (b) show box plots of the posterior marginal mean estimated by INLA for simulations of lengths $n = 500$ and
$n = 1000$, respectively. The blue line shows the true $b$ used in the simulation. Panel (c) and (d) show box plots of the estimated posterior
probability of the slope being positive given the true value for simulations of length $n = 500$ and $n = 1000$, respectively.





The results of the analysis is presented in 1 and displayed graphically as box plots in Fig. 3. Since the posterior distribution of $b$ is skewed, especially when its absolute value approaches 1, ordinary box plots would classify a larger number of points as outliers. We use instead an adjusted box plot proposed by (Hubert and Vandervieren, 2008) which is better suited for skewed distributions. We obtain decent accuracy of the posterior marginal means $\hat{b}$, with a small underestimation when $b \longrightarrow -1$ and a small overestimation when $b \longrightarrow 1$. The posterior probabilities suggests that when $|b| \geq 0.2$ there is both a low chance of false negatives (high sensitivity) and false positives (high specificity). For smaller absolute values however, especially those generated under $b = 0$, more variation in posterior probabilities were observed. This behaviour improves when $n$ increases from 500 to 1000. For $n = 500$ we find that out of $n_r = 1000$ simulations there were zero false positives for $b \leq -0.1$, and a single false negative at $b \geq 0.1$. For $n = 1000$, no false positives or negatives were found.

|  | $n = 500$ | $n = 1000$ | $n = 500$ | $n = 1000$ |
|---|---|---|---|---|
| True $b$ | $\hat{b}$ | $\hat{b}$ | $P(b>0)$ | $P(b>0)$ |
| -0.8 | -0.766 | -0.78 | 0 | 0 |
| -0.7 | -0.67 | -0.687 | 0 | 0 |
| -0.6 | -0.578 | -0.591 | 0 | 0 |
| -0.5 | -0.483 | -0.493 | 0 | 0 |
| -0.4 | -0.385 | -0.393 | 0 | 0 |
| -0.3 | -0.291 | -0.295 | 0.001 | 0 |
| -0.2 | -0.192 | -0.197 | 0.015 | 0.001 |
| -0.1 | -0.095 | -0.1 | 0.151 | 0.065 |
| 0 | -0.004 | 0.002 | 0.478 | 0.508 |
| 0.1 | 0.097 | 0.1 | 0.854 | 0.937 |
| 0.2 | 0.199 | 0.197 | 0.985 | 0.999 |
| 0.3 | 0.292 | 0.294 | 0.999 | 1 |
| 0.4 | 0.392 | 0.394 | 1 | 1 |
| 0.5 | 0.485 | 0.494 | 1 | 1 |
| 0.6 | 0.583 | 0.592 | 1 | 1 |
| 0.7 | 0.675 | 0.688 | 1 | 1 |
| 0.8 | 0.768 | 0.781 | 1 | 1 |

**Table 1.** Results from accuracy tests on $n_r = 1000$ simulated time-dependent AR(1) series of length $n$ for each $b$ ranging from -0.8 to 0.8. The table includes the ensemble average of the posterior marginal means $\hat{b}$ and posterior probabilities of positive slope $P(b > 0)$ for each value of $b$, and for time series' lengths of $n = 500$ and $n = 1000$.





## 4.2 DO-events

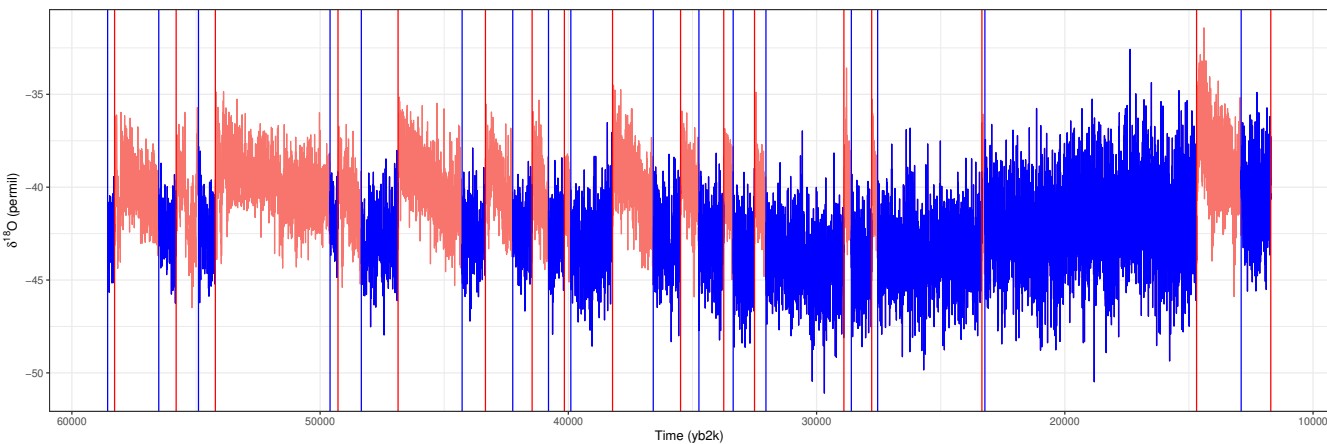

**Figure 4.** NGRIP $\delta^{18}$O proxy record. The time-series used in our study are the parts of the curves drawn in blue which are the cold stadial periods preceeding the onsets of interstadial periods drawn in red. The red and blue vertical bars represent respectively the start and the end of inter-stadial (warm) periods

We apply our time-dependent AR(1) model on the high resolution NGRIP $\delta^{18}$O record, which is partitioned into stadial and interstadial periods as shown in Fig. 4. This version of the NGRIP record is sampled regularly every 5cm steps in depth, but is non-constant in time. Having modified our model to allow for irregular time points we are able to use the raw NGRIP record without having to perform interpolation or other types of pre-processing, such as that of Boers (2018). This grants us a larger dataset for each event which could significantly improve parameter estimation. Having implemented the model using INLA we are able to take advantage of this extra resolution while keeping computational time low.

Some of these datasets appear non-stationary and thus require trend estimation. Since there is no obvious choice of forcing we consider different alternatives for trend components which are compared. The R-INLA framework allows us to very easily incorporate these trends into our model and estimates all model components simultaneously. First, we fit our model to the data without any additional trend, then we assume a linear trend, followed by a 2nd order polynomial trend. Finally, we model the trend using a continuous 2nd order random walk (RW2) spline. More details on the comparison between the different trends are included in appendix A, which also includes a plot of how well each trend fit the data.

Having looked at the fits for each event we observe that most events can be fitted easily with linear or even constant trend, but a few events require non-linearity. We choose the 2nd order polynomial trend as this gives a nice trade-off between flexibility and simplicity and appears to provide a decent fit for all events. The $\phi(t) = a + bt$ evolutions for all events using 2nd order polynomial detrending is included in Fig. 5.

The models are fitted to the stadial period preceding each of the 17 DO events and the posterior probability of $\phi(t) = a + bt$ being increasing, $P(b > 0)$, is compared for all events and trend assumptions. These are included in table 2. Using the conventional threshold of $P(b > 0) \geq 0.95$ we are able to detect early warning signals in 4 events using no detrending, and 5



events using linear or 2nd order polynomial trends and 6 events using the continuous RW2 model as a trend. Averaging over all events we are not able to conclude that early warning signals has been found over the ensemble of events for any detrending model.

Having found EWS in multiple stadial periods preceding DO events therefore indicates that DO events are not solely noise-induced unlike the hypothesis formulated in Ditlevsen and Johnsen (2010). These differences in results can be explained by
both the use of a higher-resolution dataset and a methodology not involving time windows. However, the absence of EWS in the ensemble of events does not support the hypothesis that all DO events are bifurcation-induced and hence cannot exclude the possibility for some events to be noise-induced. Our results do, however, suggest that some specific transitions may be bifurcation induced, which is in line with the results of Rypdal (2016) and Boers (2018), in which significant EWS have also been found only for some specific events. These studies use different versions of the NGRIP record from our study and their
methodologies differ from ours as they use a scale-invariant fGn model to describe the noise, as opposed to an AR(1) process.





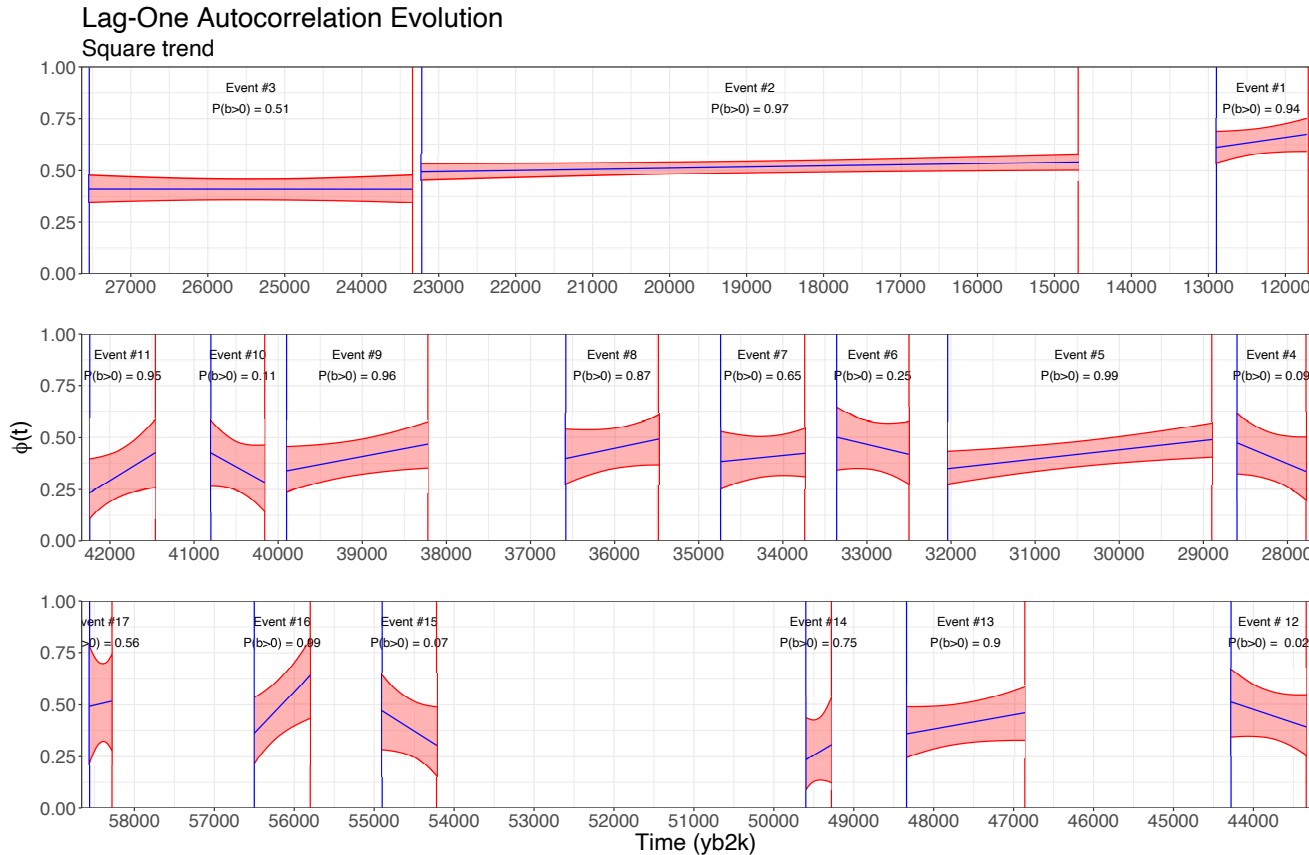

**Figure 5.** The evolution of the lag-one autocorrelation parameter $a + bt$ for each of the 17 transitions analyzed in this paper. The blue lines represents the posterior marginal means of each Greenland stadial phase, and the red shaded areas represent the 95% credible intervals. The $\delta^{18}O$ proxy measurements have been detrended using a second order polynomial. The probability of an increasing slope, $P(b > 0)$, given the posterior distribution, is also included.





| Event | No trend | Linear | Square | RW2 | Rypdal | Boers |
|---|---|---|---|---|---|---|
| 1 | 0.8824 | 0.8558 | 0.9022 | **0.9512** | $p = 0.02$ | − |
| 2 | **0.9660** | **0.9886** | **0.973** | **0.9655** | $p = 0.008$ | $p < 0.05$ |
| 3 | 0.4949 | 0.4983 | 0.5005 | 0.6137 | − | − |
| 4 | 0.0714 | 0.0878 | 0.0821 | 0.0928 | − | $p < 0.05$ |
| 5 | **0.9958** | **0.9956** | **0.9952** | **0.9918** | $p = 0.13$ | − |
| 6 | 0.2924 | 0.3052 | 0.2159 | 0.2249 | − | p<0.05 |
| 7 | 0.7569 | 0.7182 | 0.6695 | 0.9819 | − | − |
| 8 | 0.9117 | 0.9141 | 0.8747 | 0.8394 | − | − |
| 9 | **0.9862** | **0.9669** | **0.9563** | **0.9557** | $p = 0.16$ | − |
| 10 | 0.0415 | 0.1549 | 0.0942 | 0.1311 | − | − |
| 11 | 0.9325 | **0.9516** | **0.9614** | 0.9488 | − | $p < 0.05$ |
| 12 | 0.1393 | 0.1441 | 0.1268 | 0.0287 | − | − |
| 13 | 0.8898 | 0.8864 | 0.8923 | 0.9059 | $p = 0.39$ | $p < 0.05$ |
| 14 | 0.7261 | 0.866 | 0.684 | 0.7511 | − | $p < 0.05$ |
| 15 | 0.0304 | 0.0599 | 0.0695 | 0.0754 | − | $p < 0.05$ |
| 16 | **0.9903** | **0.9918** | **0.9935** | **0.9953** | − | − |
| 17 | 0.5889 | 0.5581 | 0.5766 | 0.5285 | − | − |
| Ensemble | 0.6292 | 0.6437 | 0.6216 | 0.646 | − | − |

**Table 2.** Table comparing the probability of positive slope $P(b > 0)$ for each event given posterior distributions obtained using the time-dependent AR(1) model. We ran the model using different trends including no trend (except for the intercept), a linear effect, a second order polynomial and a 2nd order random walk spline. Our results are also compared with the $p$ values obtained from Rypdal (2016) and Boers (2018).

## 5 Conclusions

This paper presents a Bayesian framework to analyze early warning signals, using an AR(1) process where the lag-one correlation parameter is assumed to increase linearly over time. Bayesian inference is obtained using a latent Gaussian model formulation and implemented using the `R-INLA` framework. In addition to computing the posterior marginal distribution for all variables and parameters in the model, implementation in the `R-INLA` framework grants a number of benefits. First, it provides a great reduction in computational cost, both in terms of speed and memory. Second, the framework is very versatile and other model components such as trends can be easily added to the predictor. Third, `R-INLA` uses posterior prediction to impute missing data automatically. The model has been applied to simulated data and shows decent accuracy.



To detect early warning signals of DO events we have applied our model to the raw 5cm NGRIP water isotope record. This record is sampled evenly in depth, but not in time requiring us to make some necessary modifications to allow for non-equidistant time steps.

Using the time-dependent AR(1) model we were unable to detect statistically significant EWS for the ensemble of 17 DO events, and only detected EWS individually for 5 events using a second-order polynomial detrending. Unlike Ditlevsen and Johnsen (2010), we find evidence of EWS in some events, corroborating Rypdal (2016) and Boers (2018). We were, however, unable to conclude that DO events are generally bifurcation-induced. To better compare with Rypdal (2016) and Boers (2018), we would have liked to employ a long-range dependent process such as the fGn. However, this task is more difficult than for the AR(1) process, as necessary modifications have to be made to the model. Moreover, this would also require working with non-sparse precision matrices which are far more computationally demanding. We did attempt to implement the time-dependent fGn model presented by Ryvkina (2015), but we were unable to ensure sufficient stability. This is, however, a very interesting topic for future work.

Currently, our model can only fit an AR(1) process where the lag-one correlation parameter is expressed as a linear function, which is not realistic. Although this is sufficient for detecting whether or not there has been a statistically significant increase in EWS, our model is unable to perform predictions or give an indication of when the tipping point could be reached. More advanced functions for the evolution of the lag-one correlation parameter should be possible, but would have to be implemented. One possible extension would be to formulate a model where the memory parameter follows a polynomial $\phi(t) = a + bt^c$, where the exponent term $c > 0$ is an additional hyperparameter. This would perhaps help give an indication of the rate of which the correlation has increased. However, when adding more parameters one needs to be careful to avoid overfitting.

To make the methodology more accessible we have released the code associated with this model as an `R` package titled `INLA.ews`. This package performs all analysis and includes functions to plot and print key results from the analysis very easily. Although this paper focuses on the detection of EWS in DO events observed in Greenland ice core records, our methodology is general and the `INLA.ews` package should be applicable to tipping points observed in other proxy records as well. We have also implemented the option of including forcing, for which the package will estimate the necessary parameters and compute the resulting forcing response. The package is demonstrated on simulated data in the appendix.

*Code and data availability.* The code and data sets used for this paper is available through the `R`-package, `INLA.ews`, which can be downloaded from: `github.com/eirikmn/INLA.ews` (last access **day month year**.

**Appendix A: Comparison of different detrending approaches**

Since there is no clear choice of forcing for DO events, and not all data windows appear stationary, we assume that there is some unknown trend component reflected in the data. This trend needs to be managed or the estimates of other components will suffer. Often, this is done by first detrending the data, before the parameters of interest are estimated. This risk that



variation caused by the time-dependent noise component may be attributed to the trend, and it is therefore better to estimate both the trend and noise components simultaneously. This can be achieved using INLA, which supports many common model components. We perform the same analysis on the data windows preceding all 17 DO events using four different trend models.

- No trend: The data is explained using the time-dependent AR(1) noise component $\varepsilon_t$ and an intercept $\beta_0$ only,

$$y_t \sim \beta_0 + \varepsilon_t. \tag{A1}$$

We only expect this to provide accurate results for stationary data windows. The results in this paper can be recreated using the INLA.ews package. Let `y` denote the $\delta^{18}$O ratios and `time` denote the GICC05 chronology, then the model can be fitted by

```
results = inla.ews(data=y, timesteps=time, formula = y ~ 1)
```

To omit the intercept term set the formula argument to `formula = y ~ -1` instead. The `rgeneric` model component corresponding to the time-dependent AR(1) noise is added automatically.

- Linear trend: We incorporate an additional linear effect $\beta_1$ in the model,

$$y_t \sim \beta_0 + \beta_1 t + \varepsilon_t. \tag{A2}$$

This can capture linear increases, but will not be able to model any non-linearity in the model. This model can be fitted using

```
results = inla.ews(data=data.frame(y=y, trend1=time_norm),
        timesteps=time, formula = y ~ 1 + trend1)
```

where `trend1 = time_norm` is the covariate corresponding to the normalized time steps,

```
time_norm = (time-time[1])/(time[n]-time[1])
```

- 2nd order polynomial: We add another effect $\beta_2$ which allows for non-linearity to be described using a second order polynomial trend,

$$y_t \sim \beta_0 + \beta_1 t + \beta_2 t^2 + \varepsilon_t. \tag{A3}$$

This model can be fitted using

```
results = inla.ews(data=data.frame(y=y,trend1=time_norm,trend2=time_norm^2),
        timesteps=time, formula = y ~ 1 + trend1 + trend2)
```



where `trend2` specifies a linear response to the covariates defined as the square of the normalized GICC05 chronology `trend2=time_norm**2`.

   – 2nd order random walk (RW2): We use a random effect $f(t)$ described by a continuous 2nd order random walk to describe the trend,

$$y_t \sim f(t) + \varepsilon_t. \tag{A4}$$

This is a continuous extension (Lindgren and Rue, 2008) of a stochastic spline model which assumes that the second-order increments are independent Gaussian processes

$$x_i - 2x_{i+1} + x_{i+2} \sim \mathcal{N}(0, \sigma_{\mathrm{RW2}}^2). \tag{A5}$$

This model is able to capture more general non-linearities compared to the 2nd degree polynomial trend, but makes the model less interpretable. Similar as in `R-INLA`, the RW2 model is specified using the following call

```
results = inla.ews(data=data.frame(y=y, idx=time,
         timesteps=time, formula = y ~ 1 + f(idx, model="crw2"))
```

where `idx` specifies the time steps of the continuous RW2 trend.

In Table 2 we present the estimated posterior probability of a positive trend, $P(b > 0 \mid \boldsymbol{y})$, compared to the corresponding $p$-values by Rypdal (2016) and Boers (2018). We show the fitted trends for each data interval in Fig A1. We observe that the

375 models tend to agree, with some exceptions where the assumed trend is unable to capture the variation of the data. Although the RW2 trend is the more flexible model it appear to exhibit irregular fluctuation for several events. The second order polynomial trend appear to be sufficiently flexible for all events, and provides a much smoother and more interpretable fit.

## Appendix B: Demonstration of the `INLA.ews` package

We demonstrate the `INLA.ews` package on simulated forced data with non-equidistant time steps. The time steps $t_k$ are

380 obtained by adding Gaussian noise such that $\tilde{t}_k = k + \xi_k$ and normalized $t_k = (\tilde{t}_k - \tilde{t}_0)/(\tilde{t}_n - \tilde{t}_0)$ such that $t_0 = 0$ and $t_n = 1$. We assume a time dependent AR(1) process of length $n = 1000$ for the observations, sampled at times $t_1, ..., t_n$. The AR(1) process has standard deviation $\sigma = 5$ and time-dependent lag-one correlation $\phi(t) = a + bt_k$ given by $a = 0.3$ and $b = 0.2$. We also include a forcing $F(t)$, obtained by simulation from another AR(1) process with unit variance and lag-one correlation $\tilde{\phi} = 0.95$. The forcing response is approximated by

$$\mu(t_k) = \frac{\sigma_f}{\sqrt{2\lambda(t_k)}} \sum_{s=t_0}^{t_k} e^{-\lambda(t_k)(t_k - t_s)} \big( F_0 + F(s) \big), \tag{B1}$$

with parameters set to $\sigma_f = 0.1$ and $F_0 = 0$, and added to the simulated observations. The AR(1) model and forcing z sampled at time points `time` can be fitted to the data y with INLA using the `inla.ews` wrapper function:





**Figure A1.** $\delta^{18}O$ proxy data from the NGRIP record (gray), with Greenland stadial phases highlighted. The posterior marginal mean (blue) and 95% credible intervals (red) of the fitted trends are included for each event.





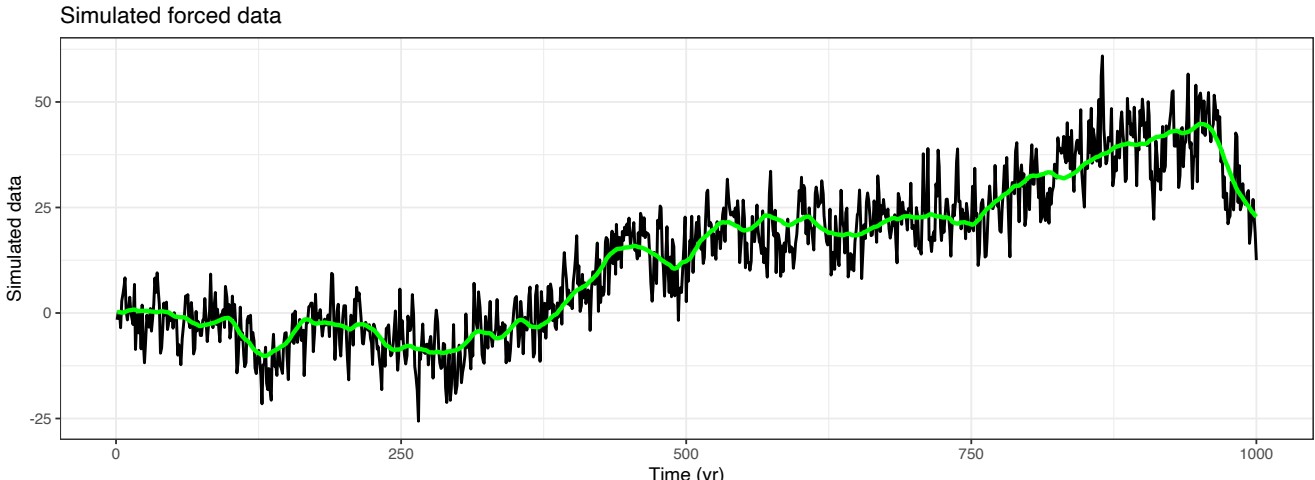

**Figure B1.** Simulated time-dependent noise (black) where a response to known forcing (green) has been added.

```
results <- inla.ews(data=y, forcing=z, formula=y ~ -1, timesteps=time)
```

The `inla.ews` function computes all posterior marginal distributions, computes summary statistics, formats the results and returns all information as an `inla.ews` list object. Summary statistics and other important results can be extracted using the `summary` function:

```
> summary(results)

Call:
inla.ews(data = y, forcing = z, timesteps = time, formula = y ~ -1)

Time used:
    Running INLA Post processing         Total
       616.7390         142.4620      759.6259

Posterior marginal distributions for all parameters have been computed.

Summary statistics for using ar1 model (with forcing):
          mean     sd 0.025quant 0.5quant 0.975quant
a       0.2938 0.0353     0.2279   0.2927     0.3672
b       0.2127 0.0449     0.1350   0.2087     0.3091
sigma   7.1593 0.3223     6.5160   7.1651     7.7789
```





| Parameter | True value | Posterior marginal mean | 95% credible Interval |
|---|---|---|---|
| $a$ | 0.3 | 0.294 | (0.228, 0.367) |
| $b$ | 0.2 | 0.213 | (0.135, 0.309) |
| $\sigma$ | 5 | 7.159 | (6.516, 7.779) |
| $\sigma_f$ | 0.1 | 0.102 | (0.091, 0.113) |
| $F_0$ | 0 | -0.004 | (-0.045, 0.034) |

**Table B1.** Underlying values used for simulating the data, along with estimated posterior marginal means and 95% credible intervals for all hyperparameters.

```
sigma_f  0.1019 0.0055     0.0910   0.1019      0.1127
F0       -0.0036 0.0202    -0.0453  -0.0028     0.0338
```

```
Memory evolution is sampled on an irregular grid.
Summary for first and last point in smoothed trajectory (a+b*time):
          mean      sd 0.025quant 0.5quant 0.975quant
phi0[1] 0.2938 0.0353     0.2279   0.2927     0.3672
phi0[n] 0.5060 0.0366     0.4370   0.5050     0.5798
Mean and 95% credible intervals for forced response have also been computed.

Probability of positive slope is 0.9999925

Marginal log-Likelihood: -3090.02
```

The results may be displayed graphically using the plot function:

```
> plot(results)
```

For this example the estimated memory evolution and forcing response is included in Fig. B2. The estimated parameters are summarized in Table B1.

Combining forcing with irregular time steps requires more computationally intensive calculations within `rgeneric`, which increases the total computational time to around ten minutes, compared to 10 seconds using any other model configuration. To reduce this we have implemented the model in `cgeneric` which grants a substantial boost in speed. However, this requires pre-compiled `C` code using more simplistic priors for the parameters, which cannot be changed without recompiling the source code. Thus there could potentially be a small loss in accuracy of the fitted model at the cost of the improved speed. To use the `cgeneric` version of the model, set `do.cgeneric=TRUE` in the `inla.ews` function call.





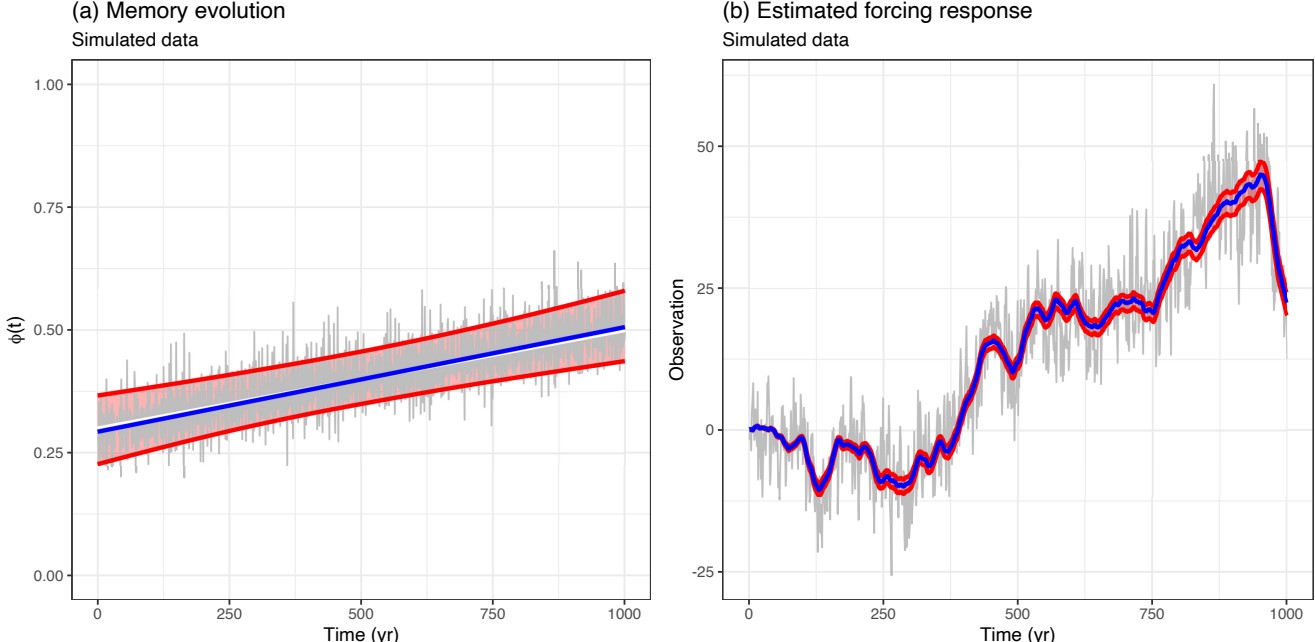

**Figure B2.** Panel (a) shows the posterior marginal mean of the lag-one correlation parameter of the simulated data (gray). The fluctuations are caused by being sampled at non-constant time steps. The posterior marginal mean of the smoother evolution of $a + bt$ is included (blue), along with 95% credible intervals (red) and the true values (white). Panel (b) shows the simulated observations (gray) along with the posterior marginal mean (blue) and 95% credible intervals (red) of the forcing response.

*Author contributions.* All authors conceived and designed the study. EMN adopted the model for a Bayesian framework and wrote the code. LH and EMN carried out the examples and analysis. All authors discussed the results and drew conclusions. EMN and LH wrote the paper with input from MWR.

*Competing interests.* The authors declare that they have no conflict of interest

*Acknowledgements.* This project has received funding from the European Union's Horizon 2020 research and innovation programme (TiPES, grant no. 820970). EMN has also received funding from the Norwegian Research Council (IKTPLUSS-IKT og digital innovasjon, project no. 332901). We would like to thank Niklas Boers for helping us reproduce the results of Boers (2018), including providing code to obtain the interpolated 5-year sampled NGRIP/GICC05 data set.



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
