# Peer review of "Bayesian analysis of early warning signals using a time-dependent model"

_EGUsphere, 2024_

## Author Comment (AC1)

General comments

The ms 'Bayesian analysis of early warning signals using a time-dependent

model' interprets geoscience time series containing DO events through the lens of an AR1 process and assumes a low-order Taylor expansion time-dependent propagator. The propagator's parameters are determined through Bayesian learning from the time series' segments that are quasi-stationary. From the analysis, the authors identify some of the DO events as bifurcations, while others are seen as merely noise-induced.

To the best of my knowledge, utilizing a time-dependent AR1 process to diagnose a system approaching a bifurcation is novel. By definition, data are exploited best by utilizing a statistic that explicitly contains the time-dependence, as against utilizing a moving window in combination with a static AR1 process. Here the ms provides a great service to the geoscience community in demonstrating such an approach can be implemented. I very much would like to see this ms being published with a 'peer-reviewed' status in an EGU journal.

Answer: We thank the referee for the helpful review. The comments will be addressed point-by-point below.

However, the ms should be modified in two main respects. Firstly, it should become clearer what is the domain of applicability of the utilized method. When analyzing time series through the lens of an AR1 process, one lives in a quadratic approximation of the potential as shown in Fig1. This in turn can only be justified in a small noise expansion. However, the ms provides no hint why the small noise expansion might be justified. Quite the contrary, the Conclusions section suggests that a subset of DO events is rather noise- than bifurcation-induced. So, the noise level is seen as large enough to trigger jumping to another equilibrium. This raises doubts whether a small noise expansion is compatible with the time series at hand.

Answer: We will add a comment addressing this in the revised manuscript.

Secondly, I would expect that most readers of EGUsphere are no trained statisticians. Most natural scientists might have heard of Bayes' formula. However, it might be useful to recover the Bayes' principle in a short Appendix. I personally found some of the wording on page 8 inaccessible, such as 'latent field'. I find it necessary that anything transcending elementary Bayesian learning is clearly defined somewhere – either in the main text (preferred) or an Appendix. So here I am asking for a didactical upgrade of the statistical method used in view of a natural science audience.

Answer: We will rewrite this section to include a high-level discussion on Bayesian inference, with more details in the appendix.

Overall, a timely and exciting to read article which is apparently on a very high technical level.

Technical corrections

1. P4: On the history of early warning systems, the following additions might be in order. (1) First mentioning of a noise-induced precursor of a bifurcation: Wiesenfeld (1985), (2) first extension to a complex system, justifying 1D AR1: Held and Kleinen (2004), (3) first utilization on real data (in fact, ice core data): Dakos et al. (2008).

Answer: These additions are appreciated and will be added to the manuscript.

2. L90-100: How does this § relate the other parts of the ms? Later on, an AR1 process is utilized, while this § seems to suggest that it should not be utilized. Furthermore, the Green's function as of Eq9 might be interpreted as a superposition of Green's functions as of Eq7 which could easily occur in multi-dimensional systems. Should one then simply expand the presented formalism to larger dimensions than one? So, I am confused about the logical positioning of that § in the overall ms.

Answer: We agree that the discussion of fractional Gaussian noise and the Hurst exponent is tangential and not really relevant to the main message of the paper which focuses on the AR(1) process. We will remove this in the revised manuscript.

3. I have issues following the rationale of Section 3.1. In the context of Bayesian learning, I would expect that an observation variable is defined (is it x or is x observed through an additive layer of noise – please clarify), and the conditional probability of an observation (in our case a time-correlated time series) on uncertain parameters is presented. In this context, I do not understand the definitions of eta, beta, z, y, for what reason I need to postpone my review of that part to another iteration.

Answer: Section 3.1 addresses how we fit our model into a hierarchical Bayesian modeling framework. Essentially, the data (y) follows a regression model expressed by the predictor (eta) which may depend on some fixed effects (beta_i) of known covariates (z_i) and noise (epsilon), here a time dependent AR(1) process. This section includes details which may be

overly technical. Moreover, we used x to denote all random variables included in the predictor (as well as the predictor itself) while in the previous section we used x to denote the time dependent AR(1) process.

We understand the confusion and will update our notation to be more consistent in the revised manuscript. We will also rewrite this section to attempt to make it easier to follow

4. Eq29: Why do we need non-constant time-steps? Is it due to missing data?

Answer: In certain applications, one may not have data sampled at constant time steps. For example, the NGRIP/GICC05 dataset used in this paper is given in constant 5cm steps in ice core depth, but the corresponding time is irregular.

5. Caption of Fig5: What are the intervals showing? Are they 2.5-97.5% quantiles of the posteriors of those variables? Or are they confidence intervals in the frequentist's sense? Would the latter logically consistent with a Bayesian setup?

Answer: The intervals are indeed the 95% credible intervals taken between the 2.5 and 97.5% quantiles of the posterior distribution. We will mention this in the main text.

6. L110: Why are you utilizing a Bayesian approach at all? You emphasize the benefit of having a PDF as output, which has tremendous advantages when later being utilized in economic decision theory. However, you do not mention the drawback of the Bayesian approach: That you need to justify the choice of a prior. What is your justification and what prior was chosen? Is it a 'vague' prior (L185, whatever that means). In the whole ms, I could not find a single reason why a Bayesian approach as against a frequentist's approach was necessary. The utilized likelihoods could likewise have been utilized for a frequentist's approach. Hence, some more motivation of the choice would be helpful.

Answer: The Bayesian framework is very useful in detecting early warning signals since we get uncertainty quantification expressed by the posterior distributions. It also allows for prior knowledge to be incorporated through the prior distribution.

The choice of prior distribution is an essential aspect of Bayesian analysis and should have been given more attention in our manuscript. Since we do not have prior knowledge about the parameters we use vague Gaussian priors, i.e. prior distributions with very large variance on all parameters. In our revised manuscript we use instead a gamma distribution for the precision $\kappa=1/\sigma^2$, and uniform priors on $a$ and $b$. Since $a$ depend on $b$ we assign a conditional uniform prior on $a$. We will discuss this in more details in the revised paper. Moreover, we will discuss how robust our model is with respect to how informative our priors are.

Specific comments

1. L168: stochastic -> probabilistic? (to distinguish a situation from aleatoric uncertainty?)
2. Define 'hierarchical Bayesian model'.
3. Define kappa before it is utilized in Eq23.
4. Activate 'day month year' (eg L330, to be found more than once in the ms).

Answer: These comments will all be addressed in the revised manuscript

Literature

Dakos, V.; M. Scheffer; E.H. Van Nes; V. Brovkin; V. Petoukhov; and H. Held. 2008. Slowing down as an early warning signal for abrupt climate change. *Proceedings of the National Academy of Sciences* 105:14308-14312.

Held, H. and T. Kleinen. 2004. Detection of climate system bifurcations by degenerate fingerprinting. *Geophysical Research Letters* 31:L23207.

Wiesenfeld, K. 1985. Virtual Hopf phenomenon: A new precursor of period-doubling bifurcations. *Physical Review A* 32:1744.

---

## Author Comment (AC2)

**Review: Bayesian analysis of early warning signals using a time-dependent model**

**General Comments:**

The manuscript 'Bayesian analysis of early warning signals using a time-dependent model' presents a new variation of critical slowing down detection also known as early warning signal analysis. The authors consider a standard tipping scenario wherein a one-dimensional random dynamical system undergoes an abrupt shift from one regime to another. Both regimes correspond to the system's residency in different basins of attractions. The basins are associated with stable equilibria of the underlying deterministic drift. Two possible way for such regime shifts to occur are either when the random fluctuations push the system across the basin boundary or when the momentarily attracting stable fixed point is annihilated in a dynamic bifurcation. A growing body of literature is concerned with anticipating catastrophic dynamic bifurcations before they occur and so is the present manuscript.

Based on a standard linearization around the attracting equilibrium the authors suggest an AR1 model for the dynamics of the system. They incorporate a potential destabilization of that equilibrium by introducing a linear trend in the autocorrelation parameter. The three parameters of this simple model may then be fitted to any given univariate time series in a Bayesian sense. The authors suggest to use the marginal posterior distribution of the trend's slope as an indicator for whether or not the system under study is undergoing a destabilization. A marginal posterior probability of more than 0.95 for a positive slope is considered an early warning indicator for a forthcoming dynamical bifurcation and therewith a nearing tipping point.

As a proof of concept the authors apply this methodology to synthetic data, where the trend is known. They find overall good agreement between the marginal posterior mean of the slope parameter and its true value. Furthermore, they attest a generally good performance of their method in detecting true positive or negative trends.

Subsequently, the apply their method to the NGRIP d18o record, which shows abrupt transitions from low to higher values. These transitions are known as the famous Dansgaard—Oeschger (DO) events, abrupt warming events that seized the high latitude North Atlantic repeatedly during the last glacial interval. The authors find early warning signals for 5 out of the 17 studied DO events under application of a suitable detrending scheme. They conclude that these DO events were primarily caused by a destabilization of the prevailing climate attractor and suggest to regard them as bifurcation induced tipping events. However, the remaining DO events for which no early warnings are found, are suggested to have been primarily noise-induced.

As far as I know, the suggested method for assessing early warning signals has not been considered yet and it complements the existing toolkit nicely. It is indeed quite elegant and circumvents the long-lasting debate on how to test significance of trends found in other early warning indicators that are commonly

assessed in sliding windows on the time series under study. If one accepts the standard although fairly heavy assumption that a destabilization of a climate state can be represented by a one-dimensional Ornstein—Uhlenbeck process with decreasing restoring force, then the mathematical derivation presented in this manuscript is sound and convincing.

Answer: We thank the referee for a helpful and thorough review. The comments will be addressed point-by-point below.

The authors already emphasize that a linear ramp of the autocorrelation is only the first step along the way. However, I would have thought that introducing a breakpoint, where the autocorrelation changes from a constant to a linear ramp, would have been possible already in this initial paper on the method. Also, I am missing a discussion on the method's robustness against changing noise levels. In the section where the method is tested on synthetic data, I believe the authors missed to specify the noise level they used to generate the AR1. I wonder if the performance was equally good, if the noise level was increased. Finally, it would be interesting to see how the method performs in the presence of time-dependent (say increasing) noise. But this might admittedly go beyond the scope of this manuscript.

Answer:

Including a break-point would help make the model more realistic and could possibly help identify early warning signals in data for which critical slowing down is only occurring in a relatively small subset of the data set. In the revised manuscript we will include a brief discussion of this extension along with a simple demonstration on simulated data.

In the robustness test we used sigma=1. Since the noise amplitude does not affect the structure of the correlation function, the estimates of $a$ and $b$ should be similar under different noise values. Having since performed the same test for sigma=10, we can confirm that we obtain comparable results both for E($b$) and P($b$>0). We will update the text to include this detail.

Adding a time-dependent noise, e.g. using a time-dependent AR(1) process, is certainly an interesting extension. One that would hypothetically provide more flexibility and predictive power to the model. However, since our focus in this paper is limited to detecting early warning signals, rather than providing a realistic model for prediction or uncertainty quantification, we feel this extension might go beyond the scope of this paper. It is, however, a very interesting possible extension for future work, which should include extensive analysis of the model's robustness and numerical stability.

I think the introduction could be shortened quite a bit. There are enough papers which explain the archetypal double-fold bifurcation model for a tipping element and that can simply be referenced, although I admit that this is a matter of preference. The paragraph on Rypdal (2016), however, is much too detailed. The Green's function of fractional Gaussian noise for example is completely irrelevant for the remainder of the paper.

Answer: We will attempt to trim the introduction, and we agree that discussion on the Green's function is not relevant for the topic of this paper and will remove it in the next iteration.

Furthermore, I acknowledge that the implementation of the method in INLA comes with a lot of computational benefits. On the other hand, the actually fairly simple mathematics behind the suggested

method are strongly complicated when squeezed into the INLA framework. In switching between the concrete AR1 model and the abstract formulation of a latent Gaussian model there are a few inconsistencies in the notation, which makes it hard for readers not familiar with INLA to follow. For example in equations (11-13) you use the variable x to denote the observations. Later x becomes the latent Gaussian field and the observations are labeled y. In Section 4, y is again the observations. You also introduce \beta and z to denote fixed effects and covariates. In that context, you say that the precision matrix in Eq. (22) is given by Eq. (14). However, this matrix has the size n x n and can work as a precision matrix without any further ado, if \beta and z are empty. It requires careful reading to understand all this and that only in the Appendix you further specify \beta. I wonder if the entire INLA section is needed at all? A straightforward formulation of the method in standard Bayesian statements would additionally allow non-R-users to implement the method relatively easily and shorten the manuscript by almost two pages.

Answer: We agree that the section on Bayesian inference is too long and too technical, which can mostly be attributed to explaining how to implement the model for INLA using the latent Gaussian model framework. We will rewrite this section such that it is shorter, more general and easier to follow, and defer the more technical details to the appendix. We will also address the inconsistencies in the notation.

The manuscript implicitly draws on the following categorization: Whenever a destabilization of a system's state, i.e. a significant positive trend in the autocorrelation parameter can be evidenced before an abrupt transition, the transition is considered 'bifurcation-induced'. When there is no such trend prior to a transition, the transition is deemed noise-induced. This ignores the fact, that for some of the stadials with significant positive trend, the autocorrelation at the end of the stadial is still lower than for some stadials which are best described by a constant autocorrelation. In this simple OU-process picture, the latter stadials are thus permanently closer to a potential bifurcation point than those which undergoe a continuous destabilization. I suggest to double-check the wording with respect to this issue. In my view, the suggested method is useful to detect a system's ongoing destabilization. But I think the distinction between noise-induced and bifurcation-induced transitions has to be taken with a grain of salt. Somewhat related to that, I would also be nice to read something about how this method could be used to monitor (climatic) systems that are prone to undergo critical transitions. In particular the Bayesian approach would allow for an updating scheme, every time new data are available. I suggest to add a paragraph on this question to the conclusions.

Answer: A statistically significant increasing trend is indeed indicative of the ongoing destabilization, not the current stability of the system. We will update the language to avoid mixing up these two terms.

One could indeed perform Bayesian updating to monitor climatic systems. This can be achieved by using the posterior distribution of the previous analysis as the prior distribution for the new analysis. We will, as suggested, include a paragraph on this in the manuscript.

As a minor remark, I would suggest to increase the font sizes in the figures.

Answer: This will be changed in the next revision

Last but not least, I wonder how robust your method is against changes in the prior distributions? I also could not find a numeric specification of the priors that you are using in the actual analysis anywhere in the manuscript? Could you please comment on that?

*Answer: We will include some discussion on the priors, including numeric values, and also extend our robustness analysis to include tests on the prior distribution.*

In summary, I find that the paper makes a nice contribution to ESD and deserves being published. The overall study design is solid and the results add an important perspective to the debate on EWS prior to DO events. On a technical level, however, the manuscript does not yet fulfill the criteria for publication and requires improvements. There are numerous linguistic inaccuracies and plenty of grammatical and spelling mistakes. I urge the authors to address the subsequent list of comments and to have a more rigorous round of internal reviews before resubmitting their manuscript.

*Answer: We greatly appreciate the comments below and will address them all before resubmission.*

**Specific Comments:**

l.3    Tipping points can be crossed solely by internal variation in the system or by approaching a bifurcation point where the current state loses stability and forces the system to move to another stable state.

It seems to me the sentence is semantically not correct. The verb 'forces' refers to the subject 'the current state'. Strictly speaking, I don't think it is correct to say that the current state forces the system to move to another stable state. Its the annihilation of that state.

*Answer: We changed "and" to "which" to hopefully make it more clear*

l.4    It is currently debated whether or not Dansgaard-Oeschger (DO) events, abrupt warmings occurring during the last glacial period, are noise-induced or caused by the system reaching a bifurcation point.

I think 'occurring' should be in past tense. DO events are introduced here on the flight. I would suggest to add at least sth. like 'warmings of the North Atlantic region' to make the sentence a bit more specific. Furthermore, the debate is not restricted to noise-induced vs. bifurcation induced tippings. Several authors have considered an excitation mechanism (e.g. Timmermann, 2003, Ganopolski, 2002, or Riechers, 2024) or limit cycle behavior, i.e. self-sustained oscillations (e.g. Peltier, 2014, Saha, 2015, Mitsui and Crucifix, 2017, Vettoretti, 2022, just to name a few).

If you simply write 'It is currently debated whether or not DO events are preceded by early warning signals' then you are on the save side, I would say. Maybe move the sentence further down, so that it comes after the introduction of an early warning signal.

Answer: We changed the sentence to "It is currently debated whether or not Dansgaard-Oeschger (DO) events, abrupt warmings of the North Atlantic region which occurred during the last glacial period, are preceded by early warning signals. " and moved it to after tippings points are introduced, as suggested.

l.8      To express this behaviour we propose a new model based on the well-known first order autoregressive process (AR), with modifications to the correlation parameter such that it depends linearly on time.

I don't think that you can rightfully call this model 'new'. What is probably, is your approach to estimate the temporal variation of the correlation parameter using a Bayesian setup.

Answer: We agree, "new" is removed.

l.12      Early warning signals were detected and found statistically significant for a number of DO events, suggesting that such events could indeed be caused by approaching a bifurcation point.

I assume you are referring to you own analysis here and not the one conducted by Boers (2018). In that case, I think the sentence should be written in present tense.

Answer: We rewrote the sentence in present tense: "Statistically significant early warning signals are detected for a number of DO events, which suggests that such events could indeed have been caused by approaching a bifurcation point."

l.18      If the state of a component of the climate system, by crossing some threshold in the form of an unstable barrier separating two basins of attraction, changes from one stable equilibrium to another it is said to have reached a tipping point.

I doubt that the the term 'unstable barrier' is appropriate here. Two basins of attraction are separated by a basin boundary, which may comprise unstable fixed points or saddles. But calling the 'barrier' unstable is probably misleading. Furthermore, the term barrier implicitly seems to refer to some sort of a potential barrier, a picture which is actually only valid in gradient systems, i.e. where the drift field can be written as the gradient of a potential.

Maybe you also want replace 'have reached' by 'has crossed'.

Notice, that here you implicitly define a tipping point as the noise-induced crossing of a basin boundary. This definition certainly is consistent with a preceding destabilization of the initial state caused by the approach of a bifurcation point. However, strictly speaking, it excludes purely bifurcation driven tipping events, which is to some extent contradictory to the definition in the abstract.

Answer: We replace the term "barrier" with "boundary of unstable fixed points", and "have reached" by "have crossed" as suggested. We have modified the definition of tipping points to also include bifurcation driven tipping:

"If the state of a component of the climate system changes from one stable equilibrium to another, either by crossing some threshold in the form of a boundary of unstable fixed points separating two basins of attraction or by having the initial equilibrium destabilize, it is said to have crossed a tipping point. "

l.19    Components of the Earth system has experienced tipping points numerous times in the past, leading to abrupt transitions in the climate system.

It should read 'have' experienced.

Answer: This will be fixed

l.24    These are known as Dansgaard-Oeschger (DO) events (Dansgaard et al., 1984, 1993) and are characterized by cycles where the temperature increased substantially, up to 16.5°C for single events, over the course of a few decades followed by a more gradual cooling, over centuries to millenia, back to the GS state.

1) I would suggest to include 'transitions' after 'These' for sake of clarity.
2) The sentence does semantically not seem correct to me. It currently states that the 'transitions' are characterized by 'cycles'. Maybe you can change it to 'these transitions are part of climatic cycles' or 'these transitions initialize a climatic cycles'. Maybe you also want to make two sentences out of this one for sake of readability.

Answer:
1)  This change will be implemented.
2)  We rewrote the sentence into two shorter sentences:
    " These transitions are known as Dansgaard-Oeschger (DO) events and initialize climatic cycles where the temperature increased substantially, up to 16.5 C for single events, over the course of a few decades. This is followed by a more gradual cooling, over centuries to millenia, returning to the GS state."

l.26    A total of 17 DO events (Svensson et al., 2008) have been found for the past 60 kyr before present (BP) and they represent some of the most pronounced examples of abrupt transitions in past climate observed in paleoclimatic records.

I would suggest to refer to Rasmussen (2014) when it comes to the 'official' number of DO events. Furthermore, I recommend to simply remove the 'before present'. 'Before present' usually takes 1950 as a reference date. since you already introduced the b2k notation, you should not mess it up with a slightly different notion of indicating ages.

Answer: We have changed the reference and we agree that introducing BP is redundant, as we already have introduced b2k and BP is only used here.

l.29    It is widely accepted that such transitions are associated with a change in the meriodional
        overturning circulation (MOC) (Bond et al., 1999; Li et al., 2010) causing a loss of sea
        ice in the North Atlantic.

        I don't have access to Bond, 1999. But I think both references are untypical for being
        cited as evidence for AMOC changes across DO events. Li (2010) is primarily concerned
        with the atmospheric response to sea ice removal, if I recall correctly. You might find
        Lynch-Stieglitz (2017), Henry (2016) or Menviel (2014 and 2020) more appropriate –
        even though AMOC changes have been considered even earlier then these papers.

        Also, I suggest to avoid making causal statements with respect to sea ice removal and
        AMOC changes. The causal relation between sea ice retreat and AMOC reinvigoration is
        still debated, but it seems more plausible that the first triggered the latter and not the other
        way around.
Answer: We have updated the references.  To make the causal statement on the AMOC changes
        less strong we have rewritten the last part of the sentence to "possibly caused by loss of
        sea ice in the North Atlantic.".

l.31    Some studies have found that DO events exhibit a periodicity of 1470 years (Schulz,
        2002), which have made some scientists suggest that the events have been triggered by
        changes in the earth system caused by changing solar forcing (Braun et al., 2005)

        1) … which has made …
        2) have been triggered by quasi-periodic changes in the solar forcing.
Answer: These changes have been implemented.

l.39    The behaviour around a tipping point can be analyzed by expressing the changes of the
        state-variable using a potential, wherein valleys represent the basins of attraction that are
        separated by an unstable fixed point.

        This sentence is really unclear. Being familiar with the topic, I may guess what you are
        trying to  say, but that is not what is written here, I believe.

        1) the **behavior** of what?
        2) 'around' (?) a tipping point? what is the behaviour 'around' a tipping point?
        3) what exactly means: expressing the changes of the state-variable using a potential? I
        imagine you mean that one can reduce the dynamics of a complex systems to a single
        dimension and then introduce a quasi-potential whose gradient corresponds to the
        reduced-dynamics drift function?

        You certainly should also not introduce dynamical system theory from scratch. But the
        statement needs a bit more clarity and preciseness, I would say.
Answer: We will attempt to rewrite this part to make it more clear.

l.42     spawn? what about vanish?

Answer: By spawn we meant they are created.

l.47     By assuming that a time-dependent state-variable $x(t)$ […] vary over some potential $V(x)$ with stochastic forcing corresponding to a white noise process $dB(t)$ […] then the stability of the system can be modeled using the stochastic differential equation […].

This sentence seems grammatically incorrect to me. I have also not come across the expression 'a state variable varies over a potential', but that does not mean that this expression does not exist.

l.58     It can be shown that the bifurcation points are

This expression typical suggests that a corresponding proof requires substantial effort. However, in this example, deriving the bifurcation points takes a couple of lines. I would suggest to write 'In this example the bifurcation points are'.

Answer: Since this section was more related to a specific example of a dynamical system and not really essential to the paper we have chosen to remove this part of the introduction.

fig.1     The potential over the set of state variables before, at and after the control parameter has reached the bifurcation point $\mu_2$. Panel (a) shows the potential and fixed points for some $\mu < \mu_2$, and panels (b)–(c) shows the same for $\mu = \mu_2$ and $\mu > \mu_2$, respectively. When the control parameter approaches the bifurcation point $\mu_2$, the stability of the stable fixed point $x_1$ decreases and eventually collapses at $x_1 = x_2 = -\xi/3$, leaving $x_3$ as the only (stable) fixed point.

I am a bit puzzled by the term 'set of state variables' in the caption. I would say that is the state space X and the state variable can assume elements from that state space.

[…] , and panels b and c show (without s)

Answer: These suggestions will be implemented.

l.62     The change in values and stability of the fixed points as we increase the control parameter is illustrated in the bifurcation diagram Fig. 2, which include the stable fixed points $x_1$ (lower solid curve) and $x_3$ (upper solid curve) and the unstable fixed points $x_2$ (middle dashed curve), representing the separating barrier.

1) I think it should read 'the change in value and stability'…
2) […], which includes

Answer: This will be fixed.

l.65     The diagram also includes a simulated process generated by the same potential which demonstrates how abruptly the state variable changes when the system crosses the tipping

threshold x 2, which happens before the control parameter reaches the bifurcation point μ 2 due to the diffusion term σdB(t).

    1) the reference of 'which demonstrates' is unclear
    2) I am not sure if one can say: 'the process generated by the potential'.
    3) it is not made explicit, that you simulated the process while continuously changing μ. Neither it is clear how you changed μ. I assume its a simple linear ramp. You might want to add a time axis to the top spline of figure 2.

Answer:
1) The "which demonstrates" is related to the simulated process. We will add punctuation to make this more clear.
2) Will change this to "generated by the same dynamical system"
3) The control parameter μ does indeed change linearly. We will make this more clear in the text.

l.75    This solution forms an Ornstein-Uhlenbeck (OU) process, which under discretization is a first order autoregressive (AR) process with variance $\text{Var}(x_t) = \sigma^2/(2\lambda)$ and lag-one autocorrelation parameter $\phi(t) = \exp(-\lambda)$.

There is a $\Delta t$ missing in the $\phi(t)$. The addition 'with the variance' could already be placed right behind 'OU process', because the variance does not follow from the discretization. I would suggest to include the formula (8) right behind the term 'autoregressive process'.

Answer: We agree with these modifications and will implement them in the revised manuscript.

l.78    When the control parameter approaches a bifurcation point we expect increased variance and correlation, as could be observed in Fig. 2.

If you go into this level of detail and introduce the linearization explicitly, I feel like you should mention, why you variance and autocorrelation increase, namely, because the \lamda – the restoring or damping force – goes to zero. Strictly speaking, I think you shouldn't use the formulation 'we expect' in this context. Variance and Autocorrelation are defined as ensemble averages and in the simple setup you consider they DO increase, it's not a matter of expectation.

Answer: The sentence will be changed to "When the control parameter approaches a bifurcation point the restoring rate $\lambda$ goes to zero and consequently the variance and correlation of the state variable will increase" to reflect this.

l.81    In fact, recent studies have discovered that more components in the earth system exhibit EWS and are at risk of approaching or have already reached a tipping point.

'more' is a comparative. Maybe you want to say 'several'.

Answer: This suggestion will be incorporated in the next version.

l.84    Analysis of EWS for DO events in the high-dimensional Greenland ice core record has
        been conducted by others, e.g. Ditlevsen and Johnsen (2010) whom applied a Monte
        Carlo approach to detect increased variance and autocorrelation in a system driven by
        white noise.

        1) What is meant by 'high-dimensional'
        2) who – not whom
        3) the addition 'a Monte Carlo approach to detect increased variance and autocorrelation
        in a system driven by white noise' is so vague / unspecific, that is does not convey any
        valuable information about their chosen methodology. From a short read, I a actually
        under the impression that Ditlevsen (2010) do not specify a null-model to test the
        significance of the early warning indicator time series they compute from the NGRIP
        data. It seems to me that the Monte Carlo sampling used to construct a null distribution is
        only used in the assessment of the synthetic data.

Answer:
    1)  "High-dimensional" was meant to refer to the fact that the NGRIP record is relatively
        large. Since this is not very relevant here we choose to remove it
    2)  This will be fixed
    3)  This will be rewritten to: "who estimated the variance and autocorrelation over a sliding
        window where the system was assumed to be driven by white noise."

l.96    Rypdal (2016) was able to detect an increase of variance of the high-frequency
        fluctuations for the ensemble average of the 17 DO events at a 5% significance level, and
        individually for five separate events.

        My interpretation of the above statement – and in particular of the term 'ensemble
        average' –  would be that Rypdal took averaged the data from 17 DO events to obtain one
        archetypal smooth transition. I assume that this interpretation is incorrect, since it would
        hardly allow for the detection of early warning signals. Can you think of another
        formulation which conveys clearer the approach followed by Rypdal?

Answer: An important assumption of Rypdal (2016) is that the system can be expressed using a
        long-range dependent process. The details on this is not really relevant to the topic of our
        paper where we focus exclusively on the AR(1) process. As such we decided to remove
        this part for the next iteration of the manuscript.

l.97    These results were corroborated by Boers (2018) whom applied a similar strategy […]

        who applied

        to the higher resolution of the NGRIP δ 18 O data set.

        to a higher resolved version of the NIGRIP data set.

        on which he applied interpolation to obtain time series with regular 5-year sampling steps.

I would say this piece of information is irrelevant here. There are many more preprocessing steps in the analysis by Boers and the re-sampling to 5 year resolution does not necessarily stand out. So I suggest you to either list all the steps or none with a clear personal preference for none.

Answer: These changes will be implemented, and we will remove the detail on Boer's interpolation.

l.101    Most approaches for detecting EWS in the current literature require estimation of statistical properties in a sliding window, e.g. by producing Fourier surrogates and estimating the Kendall's $\tau$ statistic for each iteration.

[please provide references]

This is not entirely clear to me. I assume, the 'statistical properties' you mention are for example variance and autocorrelation, which are estimated in sliding windows on the time series under study. How can you estimate these quantities 'by producing Fourier surrogates and estimating the estimating the Kendall's tau statistic for each iteration'? From what I know, you first need to estimate variance and autocorrelation in running windows. One possible way to assess the significance of the obtained indicator time series – so the temporal evolution of the windowed variance and autocorrelation – is to construct a null distribution of Kendall taus from Fourier surrogates: i.e. you need to compute Fourier surrogates for the entire original time series under study and apply the same windowed estimation procedure to all Fourier surrogates. Then you compute a kendal tau for the estimator time series derived from the original time series and for all estimator time series derived from the Fourier surrogates. Comparing the Kendall tau associated with the original time series with the distribution of the Fourier-surrogate based Kendal taus provides an objective significance criterion. It seems to me, that some bits of this procedure are captured in your statement, while some are not. Maybe you find a way to clarify this statement, or is my understanding of when and how to use the Fourier surrogates wrong?

Answer: To make this easier to follow, we omit the mention of Fourier surrogates and Kendall's tau statistic and instead use the more elementary statistics variance and correlation as examples.

l.102    Consequently, this presents a choice on the length of the window.

Did you mean 'this requires a choice'?

Answer: We did indeed mean that, and will update the wording accordingly.

l.119    The $\delta 18$ O ratios are frequently used in paleoscience as proxies for temperature of precipitation

Did you mean temperature and precipitation?

Answer: We meant temperature at the time of precipitation. We will update the text to make this more clear.

l.128   (last accessed: day month year)
Answer: This was an oversight and will be fixed.

l.130   During critical slowing down stationarity can no longer be assumed as we expect both the correlation and variance to increase.

Why do you say 'no longer'? Did you assume stationarity at any earlier point in the paper? Again, if CSD really takes place, then correlation and variance DO increase. The tricky part is that their statistical estimators may not increase if one has bad luck.
Answer: By stationarity we mean that the statistical properties remain constant over time. During critical slowing down the variance and correlation increases and a stationary process, such as an AR(1) process with constant lag-one correlation parameter in (0,1) will no longer be viable. Thus, we need to make adjustments, either by splitting the dataset into windows or, as we do in this section, by using a time-dependent AR(1) process.

l. 139   The time-dependent AR(1) process is expressed by the difference equation
$x_t = \phi(t)x_{t-1} + \varepsilon_t$, $\varepsilon \sim N(0, \sigma_\varepsilon^2)$, $t = t_1, ..., t_n$,       (13)
for which the covariance between two variables $x_i$ and $x_j$ is given by $Cov(x_i, x_j)$.

Isn't this already clear from Eq.(8) and Eq.(12) ?
Answer: We will remove equation (13) and instead refer to equation (8).

l.145   Is the prefactor in front of the precission matrix \sigma or \sigma_{\epsilon}? In the former case, which \sigma?
Answer: We will update the notation to make this more clear.

l.153   In fitting the model it is beneficial that the model parameters are defined on an unconstrained parameter space.

To some observational data, I assume?
Answer: Correct. We will add this to the text.

l.155   Assuming the lag-one autocorrelation parameter is defined on the interval (0, 1), and since $t \in [0, 1]$, then the slope must be constrained by

So far you didn't mention that you would rescale time, did you? Full stop behind |b|<1.
Answer: We moved the section with irregular time steps to just before Section 3.1. The rescaling of the time steps $t_k$ is described therein.

l.160   The parameter space for a depend on the current state of b

I guess what you are trying to say, is that for you model only combinations of a and b are eligible, that don't violate \phi <1 at any time. However, the use of the term 'current state of b' sound like there were some temporal changes in b, which is not the case as far as I understand.

Answer: Correct. If *b* changes, then the support of *a* will change due to the mentioned constraints.

l.164    This is strange: in this 2D parameter space the one dimension's parameterization depends on the value along the other direction. Well, actually that should pose any problems. It's just the backtransformation that depends on b itself. So the parameter space is spanned by \theta_b and \theta_a and then computing. And then b = b(\theta_b) and a = a(\theta_a, \theta_b).

Maybe this can cause some issue with the prior distribution. Say the prior was uniform, then a lot of probability density would be attributed to extreme values of a and b. Let's see how the authors solve this issue.

Answer: One concern we had was how this could affect numerical stability. However, simulation tests show that both prediction accuracy and robustness to prior selection seem to be retained. One issue with this method is that assigning priors becomes more difficult and less intuitive, since the parameters are entangled. In this paper we choose to simply assign priors on the transformed parameters \theta_a and \theta_b, though there might be better ways to do this.

l.172    Here, $\beta_0$ represent an intercept, $\beta_i$ are fixed effects corresponding to covariates $z_i$ and $\varepsilon$ are random effects representing some time-dependent noise that depend on some parameters $\theta$

represents
that depends

Answer: This will be fixed.

l.174    The covariance structure of the different components in the model are expressed by a latent field of random variables containing the predictor and all stochastic terms therein , i.e. $x = (\eta, \beta, \varepsilon)$.

is expressed

Answer: This will be fixed.

l.176    Assigning a Gaussian prior on x the model becomes a latent Gaussian model, a subset of Bayesian hierarchical models for which there exists additional computational frameworks. The latent Gaussian model is specified in three stages as follows.

I think it needs to read 'upon assigning a Gaussian prior […].'

Answer: This section will be rewritten entirely for the revised manuscript and we will keep this suggestion in mind.

l.179    The first stage is to specify the likelihood of the model.

         I don't think it is accurate to say 'the likelihood of the model'. In my view it should be the
         likelihood of the observations, given the model including its parameters.
Answer: This will be changed when we rewrite this section..

l.183    The second stage in specifying a latent Gaussian model is to specify a Gaussian prior
         distribution for the latent field x, with mean vector $\mu = E(\theta)$ and precision matrix Q.

         Did you really mean $\mu = E(\theta)$? Or rather $\mu = E(x|\theta)$ ?
Answer: We actually meant the latter. This will be fixed in the revised manuscript.

l.185    $\beta$ are assigned vague Gaussian priors and the noise term

         I think you should either write $\beta\_i$ are assigned… or **$\beta$** is assigned a vague Gaussian prior.

         What does 'and the noise term' mean?
Answer: The noise term refers to the random effects in the predictor (non-fixed effects), here that
         would be the time-dependent AR(1) process. We will address notation issues.

l.186    Specifically, for the linear predictor we assume

         $x \mid \theta \sim N (\mu, Q(\theta) -1 )$, (22)

         such that the latent variables corresponding to a potential $\beta$ component represent vague
         Gaussian priors and those corresponding to $\varepsilon$ represent the chosen model.

         To what extent does Equation (22) express something 'specific' to the linear predictor?
         Isn't this just the mathematical formulation of what you wrote in line 176: 'Assigning a
         Gaussian prior on x the model becomes a latent Gaussian model […].'

         By saying 'such that the latent variables corresponding to a potential $\beta$ component
         represent vague Gaussian priors' you mean that those components of x which correspond
         to $\beta$ are distributed according to a vague Gaussien prior? I don't think the formulation 'a
         random variable (the latent variable in your sentence) represents a distribution (a vague
         Gaussian prior)' is meaningful in general.

         Finally, I do not fully understand what is meant by 'those [latent variable] that correspond
         to $\varepsilon$ represent the chosen model'. How can some components of x represent the entire
         model?

         I am under the impression that what you did is actually correct and very sensible.
         However, your manuscript does not convey your analysis really well, or at least it takes a

lot of time to jump back and forth between the different levels and the conflicting notations: it is very tedious to identify the correspondences between the concrete AR1 model and the abstract formulation of a latent Gaussian model. E.g. the entire model defined by equations (13) and (10) is comprised in the **\epsilon** term in equation (20) if I am not mistaken. This is even more confusing, since equation (13) contains an epsilon on its own, which not equivalent with the \epsilon from equation (20), which is part of the latent field **x** and to which you also refer in line 189.

Moreover, I don't see where you need any \beta_i in the actual analysis? Could you think of a way to present your method without introducing \beta and z, which are actually superfluous?

Answer: This section will be rewritten entirely for the revised manuscript. We will then address the notation issues, the incoherent structure of the text and to keep it simple we will not discuss things that are not needed to understand the model used in this paper.

l.189    The precision matrix is given by Eq. (14).

This sentence actually requires that x from equation (22) corresponds to x in equation (11) and that \beta and z do in fact not have any entries. If my interpretation is correct, making this more explicit would help the readability of your manuscript.

Answer: Generally, \beta actually do have entries in the precision matrix, they correspond to a diagonal matrix block of the joint matrix wherein precision matrix stated in this manuscript is another block. Since the \beta variables are not essential to understand the model at hand we understand that this is confusing. We will rewrite this section and avoid mentioning \beta to make the actually important parts more clear.

l.190    The final stage concerns the prior distributions of the model parameters,

With model parameters, you mean hyperparameters? \theta_a and \theta_b are certainly not all model parameters, no matter if you refer to equation (20) or equation (13) as the model. Or is the \sigma in \kappa = 1/ \sigma² equivalent to the \sigma_\epsilon in equation (13)?

Answer: Hyperparameters are parameters of a prior distribution. When we adapt our model for the R-INLA framework we need to specify the model in a three stage latent Gaussian model. Originally (and intuitively) the likelihood is Gaussian governed by a set of parameters (not hyperparameters). To make this model fit with R-INLA we say that the Gaussian AR(1) model is actually the latent field, and the likelihood is now an iid Gaussian with negligible variance, essentially stating that the likelihood=the latent variables. It is here that the parameters \theta become hyperparameters since they are now parameters of the prior distribution. This is all very technical, unimportant to the theory and probably confusing to most readers. Hence, we will rewrite this section entirely and focus on the more simple model specification instead of the on modified for R-INLA.

l.192   For the analysis performed in this study we have assigned a penalised complexity prior
        (Simpson et al., 2017) for the scaling parameter $\kappa = 1/\sigma^2$ and Gaussian priors for the
        parameterized memory parameters $\theta_a$ and $\theta_b$.

        What is the scaling parameter and where is it used? Also, what is the \sigma in the scaling
        parameter. Does it correspond to any of the previous \sigmas?

        Are the two Gaussian priors for \theta_a and \theta_b independent? What shape does the
        distribution assume under the nonlinear transformation (\theta_a, \theta_b) => (a, b)? In
        the (a,b) plane the prior will certainly not be Gaussian anymore? What I find confusing, is
        that the transformation \theta_a => a actually depends on b (or theta_b). So, if the two
        Guassian priors were independent, then the Gaussian prior on \theta_a would transform
        differently into the space of a depending on the value of b. Could you please comment on
        that?

Answer: The scaling parameter \kappa is the precision, or the inverse of the variance. In the
        revised manuscript we will define this earlier in the manuscript. In our revision we have
        changed the prior distributions. We now use a gamma distribution on \kappa instead of a
        penalised complexity prior, and assign uniform priors on *a* and *b*. Since *a* depend on the
        value of *b* we assign a conditional prior on a | b on (*a*_lower, *a*_upper). We will describe
        this more clearly in our revised manuscript.

l.206   Instead of using simulations, INLA use various numerical optimization techniques to
        compute an accurate approximation of the posterior marginal distributions.

        INLA uses

        Accurate approximation sounds a bit odd. I assume the accuracy of the approximation
        depends on numerous factors and can probably not be guaranteed under all possible
        circumstances, can it? Consider 'appropriate approximation'.

Answer: This will be taken into account when we rewrite this section for the next version of the
        manuscript.

l.209   In line 198 you define the 'joint posterior distribution' as $\pi(x, \theta \mid y)$. Equation (27) is said
        to approximate the 'joint posterior distribution' but non of the terms inside the equation
        coincides with the 'joint posterior distribution' as introduced above. Could you please
        comment on that?

Answer: Equation (27) aims to approximate a different joint posterior distribution, namely that of
        all parameters \theta. $\pi(x, \theta \mid y)$ defines a different joint posterior distribution, which also
        includes the latent variables x. We understand that this is confusing and we will consider
        this distinction when we rewrite this section.

l.211   Maybe you could reverse the order of $\theta$ and y on the left side of the equation. Then it
        would be a bit more obvious, that this equivalence follows from Bayes Theorem.

Answer: This section will be rewritten, but we will keep this comment in mind.

l.218    (last access: day month year)
Answer: This will be fixed

l.218    Inference can then be produced by executing inla.ews(y, formula=formula), where *y* is a
        numeric vector containing the data and *formula* describes the trends included in the
        model.

        What exactly is meant by 'formula describes the trends included in the model'? Do you
        mean the possible trends of the autocorrelation parameter? I thought that one was
        constrained to a linear trend? Or do you mean any trend in the data under study?
Answer: The 'formula' argument allows the used to include different effects to the predictor.
        Instead of detrending the dataset during preprocessing one could here specify e.g. a linear
        or polynomial trend that will be estimated simultaneously as the time-dependent AR(1)
        process (which is still constrained to a linear trend, except for a few implemented
        extensions).

l.230    I would prefer $\sigma^2(t_k)$ instead of $\sigma(t_k)^2$.
Answer: We moved this section to before Section 3.1, and use this opportunity to move towards
        using the precision $\kappa$ instead og the variance $\sigma^2$, which avoids the square in
        the notation.

l.237    myrvoll-nilsen2020
Answer: This was an oversight and will be fixed in the revised manuscript.

l.237    In this subsection we adopt a similar strategy to myrvoll-nilsen2020 with changes to allow
        for time-dependence and non-constant time steps.

        Changes of what?
Answer: We modify the Green's function to allow for time-dependence and irregular time steps.
        We will mention this more clearly in the text.

l.243    where $\varepsilon(t)$ is a time-dependent OU process

        What do you mean by 'time-dependent OU process'? Judging from your equation (33-34)
        it seems that one can decompose solutions to (32) into a OU process + deterministic
        response to the forcing. But I don't see, how the OU process, which apparently describes
        fluctuations around the deterministic trajectory, should be time-dependent?
Answer: By time-dependent OU process we mean the same process as defined earlier in the
        manuscript as the restoring rate $\lambda$ changes as we approach a bifurcation point.

l.242    You might want to consider using different variable in the decomposition of x(t) to avoid
        double use of both µ and $\epsilon$. This can lead to confusion.
Answer: We agree, and we will change the notation in the next version.

l.244   Equation (33-34) seems to be a fairly general and established result. However, if you could provide a reference here, for the interested reader to understand the origin of this equation, that would be helpful.

Answer: A reference for these equations will be added

l.245   $\sigma_f^2(t) = \sigma_f^2/(2\lambda(t))$ is an unknown scaling parameter and $F_0$ is an unknown shift parameter.

What do you mean by 'unknown'? Is there no way to compute these coefficients analytically? You program seems to find these parameters somehow?

It seems you have defined neither \sigma_f nor \lambda(t) which occur both in the definition of \sigma_f(t).

Answer: By unknown we mean that the parameter value is not known and needs to be estimated as additional model parameters. \sigma_f is an additional parameter, as is F_0. The restoring rate \lambda(t) is the same as earlier and defined as log(\phi(t)), implied by equation (13). We can define it explicitly in the revised manuscript.

l.251   What is the noise level on the synthetic AR1 processes?

Answer: The noise level on the synthetic AR(1) processes is 1. We will mention this in the text.

Fig.3   Panel (c) and (d) show box plots of the estimated posterior probability of the slope being positive given the true value for simulations of length n = 500 and n = 1000, respectively.

Maybe it is clearer to say 'of the slope being positive in dependence on the true slop b'. The term 'given' in combination with some posterior probability sounds like the posterior was derived 'given' the true slope. That is not the case, the posterior was computed 'given the data' but without knowledge about the true slope.

Answer: Rewritten to: "Panels (c) and (d) show box plots of the estimated posterior probability of the slope being positive against different true values used for simulations of length n=500 and n=1000, respectively."

l.256   The results of the analysis is presented in 1 and displayed graphically as box plots in Fig. 3.

The results **are** presented. I assume you mean Tab. 1?

Answer: This will be fixed.

l.258   We use instead an adjusted box plot proposed by (Hubert and Vandervieren, 2008) which is better suited for skewed distributions.

I recommend specifying in the caption of Fig. 3 which percentiles you show in the box plots and which points you classify as outliers.

Answer: We include a simple description of the box plots. The full range of the whiskers is too technical to include here, but it presents an adjustment to the 1.5 times IQR.

l.259   We obtain decent accuracy of the posterior marginal means $\hat{b}$, with a small underestimation when $b \rightarrow -1$ and a small overestimation when $b \rightarrow 1$.

Not the other way around? It seems that the 50$^{th}$ percentile is larger than the true b for b $\rightarrow -1$ and vice versa.

Answer: Correct. We appear to have mixed these up. We are very grateful for you to have pointed this out.

Tab.1   Maybe it would be a bit more accurate to place < > around the column titles to indicate that the quantities in the table are ensemble averages.

Answer: We agree that this would make it more clear and will implement this change

l.260   The posterior probabilities suggests that when $|b| \geq 0.2$ there is both a low chance of false negatives (high sensitivity) and false positives (high specificity).

I wonder if the terms 'false negatives' and 'false positive' are meaningful in a fully Bayesian setup. What would a false negative be? For each realization of the process for a given a and b your analysis returns a posterior distribution for b, but not a point estimate which could be false negative or false positive.

Answer: By false negative we mean that we are unable to detect a positive trend under a predetermined criteria, here P(b>0), when the trend is actually positive (b>0). In our analysis we counted the false negatives incorrectly. This will be corrected in the next version of the manuscript

l.261   For smaller absolute values however, especially those generated under b = 0, more variation in posterior probabilities were observed.

… was observed. Although 'more variation' is rather unspecific here. Does it mean that individual posteriors show a larger variance at b=0 compared to b>0.2? Or that mean and shape of posterior distributions estimated for b=0 varies stronger between different realizations of the process than compared to b>0.2?

Answer: 'were' will be changed to 'was'. We mean the estimated posterior probability P(b>0) vary more when | b | <= 1. We will make this more clear in the text.

l.263   For n = 500 we find that out of n r = 1000 simulations there were zero false positives for b $\leq -0.1$, and a single false negative at b $\geq 0.1$. For n = 1000, no false positives or negatives were found.

Again, I assume you are using the posterior marginal mean as a point estimator to specify false positives / negatives?

Answer: We actually use P(b>0) to determine whether an estimate of a simulation is counted as a
false positive, true positive, false negative or true negative. We count an estimate as a
false positive if P(b>0)>0.95 when b <= 0 and a false negative if P(b>0)<0.95 when b >
0. We will make this more clear in the text.

Fig.4    inter-stadial

Interstadial
Answer: This will be fixed

l.276    [...] which also includes a plot of how well each trend fit the data.

fits the data
Answer: This will be fixed
l.278    Having looked at the fits for each event we observe that most events can be fitted easily
with linear or even constant trend, but a few events require non-linearity.

constant trend = no trend?
Answer: This would be more accurate, and will be changed in the next version.

Fig.5    I wonder if the posterior distribution of \phi(t) is actually the most interesting quantity to
show. In my view, the relevant quantity would simply be the marginal posterior
distribution of b. The current version of the figure includes the influence of the parameter
a which is irrelevant for the assessment of whether or not a destabilization of the
currently attracting equilibrium is happening during the analyzed time window.
Answer: We feel that including the posterior distribution of the memory evolution offer a great
visual indication of whether the trend is positive or not. Though we agree that really it is
the marginal posterior of b which is really the most interesting quantity. We will add the
marginal posterior distributions of b for all events, with a vertical line at b=0 to help
convey whether the trend is positive or not.

l.282    The models are fitted to the stadial period preceding each of the 17 DO events

I think it should read 'the stadial periods' in plural.
Answer: This will be fixed

l.283    These are included in table 2.

These = the results?
Answer: "These" will be changed to "the results"

l.283    Using the conventional threshold of P (b > 0) ≥ 0.95

Personally, I find the choice of this threshold reasonable, although it might be somewhat conservative. One could arguable also raise an early warning if the marginal posterior distribution assigns a probability of 75% to a destabilization (an increasing b).

Beware that you are talking about Bayesian credible intervals and not frequentist significance thresholds.

Answer: We chose to use a threshold to help summarize the results and by highlighting the events for which P(b>0) was particularly high. 95% was chosen simply because it is the most commonly used significance threshold in statistics. A strength of the Bayesian framework is that we get a complete description of the uncertainties via the posterior distribution which describe much more than whether or not an arbitrary threshold was chosen.

l.285 Averaging over all events we are not able to conclude that early warning signals has been found over the ensemble of events for any detrending model.

Which quantity do you average over all events?

Signals have been found…

Answer: We average over the estimated P(b>0) for all events which are weighted equally. We will change "has" to "have".

l.290 However, the absence of EWS in the ensemble of events does not support the hypothesis that all DO events are bifurcation-induced and hence cannot exclude the possibility for some events to be noise-induced.

What is meant by the absence of EWS in the ensemble of events? In the average (whatever the event average is)? Or does this refer to the fact that the majority of events is not preceded by EWS?

I am not sure if one can say 'the absence A cannot exclude B'? You might consider rewriting: 'Given the absence … one cannot exclude… '

Answer: What we meant was that since early warning signals were not detected in the ensemble average, i.e. the average of the P(b>0) was not found to be significant, then these results are not strong enough to suggest that all events are bifurcation induced.

l.293 in which = wherein ?

Answer: We agree and will implement this suggested change in the revised manuscript.

l.294 These studies use different versions of the NGRIP record from our study and their methodologies differ from ours as they use a scale-invariant fGn model to describe the noise, as opposed to an AR(1) process.

This is only true for Rypdal (2014) but not for Boers (2018).

Answer: We will correct this in the revised manuscript.

l.298     Bayesian inference is obtained using a latent Gaussian model formulation and implemented using the R-INLA framework. In addition to computing the posterior marginal distribution for all variables and parameters in the model, implementation in the R-INLA framework grants a number of benefits. First, it provides a great reduction in computational cost, both in terms of speed and memory. Second, the framework is very versatile and other model components such as trends can be easily added to the predictor. Third, R-INLA uses posterior prediction to impute missing data automatically. The model has been applied to simulated data and shows decent accuracy.

Probably, it is worth mentioning these aspects in the discussion? I would not consider them conclusions from your study.

Answer: We agree that this does not really fit in the conclusions. We will mention these aspects elsewhere in the paper.

l.299     In addition to computing the posterior marginal distribution for all variables and parameters in the model, implementation in the R-INLA framework grants a number of benefits.

Parameters or variables? For the variables wouldn't it be the posterior predictive distribution?

Answer: It would be the posterior predictive distribution conditioned on the actual observations. Since we in this case we do not have latent variables it works a little differently. To make the model fit within the R-INLA framework we define a set of latent variables to directly correspond to the observed values. The posterior marginals for these variables would have negligible variance and would essentially just be equal to the observations.

This implementation detail is one of the main reasons why the Bayesian inference section was so long and technical. We will omit these details when we rewrite that section as we feel it will likely confuse more than it would clarify.

l.310     To better compare with Rypdal (2016) and Boers (2018), we would have liked to employ a long-range dependent process such as the fGn.

Again, the fGn is only inherent to Rypdal as far as I know.

Answer: We will update the text accordingly.

l.316     Currently, our model can only fit an AR(1) process where the lag-one correlation parameter is expressed as a linear function, which is not realistic

as a linear function of time (for sake of clarity)

Answer: We agree. This will be changed.

l.317 Although this is sufficient for detecting whether or not there has been a statistically significant increase in EWS, our model is unable to perform predictions or give an indication of when the tipping point could be reached.

There is no such thing as significance in Bayesian statistics.
Answer: This part of the sentence will be rewritten to: "Although this is sufficient for detecting whether or not there are EWS expressed by a linear trend, …"

l.334 This risk that

This bears the risk that
Answer:  This will be implemented as suggested.

l.342 can be fitted by

Maybe: 'can be fitted by prompting'

Full stop after the behind the prompt.
Answer: This suggestion will be implemented.

l.348 This can capture linear increases, but will not be able to model any non-linearity in the model.

The model cannot capture any non-linear trend in the data, not in the model.
Answer: Correct. This will be fixed in the revised manuscript.

l.375 Although the RW2 trend is the more flexible model it appear to exhibit irregular fluctuation for several events.

is the most flexible model it appears
Answer: These will be fixed.

l.376 The second order polynomial trend appear to be sufficiently flexible for all events, and provides a much smoother and more interpretable fit.

appears
Answer: This will be corrected

l.381 We assume a time dependent AR(1) process of length n = 1000 for the observations, sampled at times t 1 , ..., t n .

> If I understand you correctly, strictly speaking you assume the process is sampled at times $\tilde{t}_1, \ldots, \tilde{t}_n$, which you then normalize for practical reasons.

Answer: Correct. It seems like there has been a mix up in the notation. We are grateful for you to point this out.

l.381 The AR(1) process has standard deviation $\sigma = 5$ and time-dependent lag-one correlation $\phi(t) = a + bt$ k given by $a = 0.3$ and $b = 0.2$.

> What do you mean by standard deviation of the process? I assume you actually refer to the amplitude of the stochastic process component? If you referred to the standard deviation of process (as you write) the amplitude of the noise would have to be time dependent to comply with standard deviation $\sigma = 5$.

Answer: Standard deviation might actually not be the best term here since the variance of x_t depend on \lambda. \sigma is in fact the the parameter used in the marginal variance Var(x) = \sigma^2/(2\lambda). We will use \kappa to make it more clear what we mean.

l.386 The AR(1) model and forcing z sampled at time points time can be fitted to the data y with INLA using the inla.ews wrapper function:

> So here you consider a setup where the forcing can actually be measured alongside the observational target variable?

Answer: Correct. We will add a comment on this where we introduce forcing earlier in the manuscript.

Fig. A1. In my view, you do not use the space optimally in this figure. All the interstadial periods are actually irrelevant in this figure. However, it would be nice to have the different trend models for the same stadial period directly below each other for better comparability.

Answer: We will attempt to change the figure to make it more easier to compare the different trends for the same intervals.

**References**

Boers, N. Early-warning signals for Dansgaard-Oeschger events in a high-resolution ice core record. *Nat. Commun.* **9**, 1–8 (2018).

Timmermann, A., Gildor, H., Schulz, M. & Tziperman, E. Coherent resonant millennial-scale climate oscillations triggered by massive meltwater pulses. J. Clim. 16, 2569–2585 (2003).

Ganopolski, A. & Rahmstorf, S. Abrupt Glacial Climate Changes due to Stochastic Resonance. Phys. Rev. Lett. 88, 038501 (2002).

Riechers, K., Gottwald, G. & Boers, N. Glacial abrupt climate change as a multi-scale phenomenon resulting from monostable excitable dynamics. *J. Clim.* 2741–2763 (2024) doi:10.1175/jcli-d-23-0308.1.

Saha, R. Millennial-scale oscillations between sea ice and convective deep water formation. *Paleoceanography* **30**, 1540–1555 (2015).

Peltier, W. R. & Vettoretti, G. Dansgaard-Oeschger oscillations predicted in a comprehensive model of glacial climate: A 'kicked' salt oscillator in the Atlantic. *Geophys. Res. Lett.* **41**, 7306–7313 (2014).

Mitsui, T. & Crucifix, M. Influence of external forcings on abrupt millennial-scale climate changes: a statistical modelling study. Clim. Dyn. 48, 2729–2749 (2017).

Vettoretti, G., Ditlevsen, P., Jochum, M. & Rasmussen, S. O. Atmospheric CO2 control of spontaneous millennial-scale ice age climate oscillations. *Nat. Geosci.* **15**, 300–306 (2022).

Rasmussen, S. O. *et al.* A stratigraphic framework for abrupt climatic changes during the Last Glacial period based on three synchronized Greenland ice-core records: Refining and extending the INTIMATE event stratigraphy. *Quat. Sci. Rev.* **106**, 14–28 (2014).

Lynch-Stieglitz, J. The Atlantic Meridional Overturning Circulation and Abrupt Climate Change. *Ann. Rev. Mar. Sci.* **9**, 83–104 (2017).

Menviel, L. C., Skinner, L. C., Tarasov, L. & Tzedakis, P. C. An ice–climate oscillatory framework for Dansgaard–Oeschger cycles. *Nat. Rev. Earth Environ.* **1**, 677–693 (2020).

Henry, L. G. *et al.* North Atlantic ocean circulation and abrupt climate change during the last glaciation. *Science (80-. ).* **353**, 470–474 (2016).

---

## Author Response (AR2)

Reviewer 1: ***Accepted as is***

Reviewer 2: ***Accepted subject to technical corrections***

I found the manuscript 'Bayesian analysis of early warning signals using a time-dependent model' by Myrvoll-Nilsen et al. to be an interesting read, especially as someone who uses these types of generic early warning signals often, and has to make decisions over the length of a window used to calculate them on. Furthermore, any method that can provide a sense of certainty through Bayesian statistics is building upon current techniques.

I realise I am seeing this manuscript for the first time after a round of revisions and after looking through the authors' responses I can see that the manuscript has been improved greatly. I only really have very minor suggestions which I have detailed below:

Generally: It is for sure a personal choice, but I would find 'autocorrelation parameter' rather than 'correlation parameter' throughout the manuscript to be easier to follow, particularly in the abstract.

Reply: We agree and will change the use of 'correlation parameter' to 'autocorrelation parameter' throughout the paper.

Generally: Don't forget to update the access dates, on the R package particularly.
Reply: These have been updated.

Figure 1: It might be worth making clear in the figure caption that for panel b, the red and black points are on top of each other in the part where you say they eventually collapse into each other. Also, here the 3rd from bottom line at the end should say 'The red line represents', the s is missing.
Reply: We will add a small comment to make it more clear that they are coinciding and we will address the missing 's'.

Figure 2: The start of the caption says 'n=500' but it is actually for n=500 and n=1000 for the figure overall so perhaps refer to these only for the specific panels later in the caption. It is worth noting that the outliers are shown as points too.
Reply: We have removed 'n=500' from the beginning of the caption, as suggested by the reviewer. We have also expanded the sentence describing outliers to mention that they are also included in the plot.

Section 4.2: Most likely as part of Figure 5, it would be good to know how many data points are used in the analysis for each event.
Reply: We appreciate this suggestion and we will add this to table 2.

Figure 5: Again personal choice but I would swap the top and bottom rows so the figure reads from earliest in the top left to latest in the bottom right, this may also mean reordering Figure 6.
Reply: We agree with this suggestion and have reordered both Figure 6 and Figure C1. We also removed the P(b>0) value from Figure 6 as it is also shown on Figure 7 where there is more space.

Line ~265: Is it possible to delve into how much agreement there is between the Rypdal and Boers results in Table 2? Clearly there is agreement with Event 2, and with Rypdal in Event 5 etc. There is a similar level of agreement with both papers, compared to the agreement the Rypdal and Boers papers have themselves. Without fully counting the instances myself, comparing these agreements across all 3 looks like it could show this method finds things that the other papers themselves disagree with on a number of occasions.

Reply: We added a comment about the agreements between these two studies and ours.

Line 375: Need an 's' on represents.

Reply: There appeared to already be an 's' on 'represents'.

Line 380: Need an 's' on depends.

Reply: Fixed.

Additional changes:

We also fixed a small error in Eq. (25), and capitalized some words.